

# Non-perturbative intertwining between spin and charge correlations: A "smoking gun" single-boson-exchange result

**Severino Adler[1,2]⋆, Friedrich Krien[1], Patrick Chalupa-Gantner[1], Giorgio Sangiovanni[2] and Alessandro Toschi[1]**

**1** Institute of Solid State Physics, TU Wien, 1040 Vienna, Austria
**2** Institut für Theoretische Physik und Astrophysik and Würzburg-Dresden Cluster of Excellence ct.qmat, Universität Würzburg, 97074, Würzburg, Germany

⋆ severino.adler@tuwien.ac.at

## Abstract

We study the microscopic mechanism controlling the interplay between local charge and local spin fluctuations in correlated electron systems via a thorough investigation of the generalized on-site charge susceptibility of several fundamental many-electron models, such as the Hubbard atom, the Anderson impurity model, and the Hubbard model. By decomposing the numerically determined generalized susceptibility in terms of physically transparent single-boson exchange processes, we unveil the microscopic mechanisms responsible for the breakdown of the self-consistent many-electron perturbation expansion. In particular, we unambiguously identify the origin of the significant suppression of its diagonal entries in (Matsubara) frequency space and the slight increase of the off-diagonal ones which cause the breakdown. The suppression effect on the diagonal elements originates directly from the electronic scattering on local magnetic moments, reflecting their increasingly longer lifetime as well as their enhanced effective coupling with the electrons. Instead, the slight and diffuse enhancement of the off-diagonal terms can be mostly ascribed to multiboson scattering processes. The strong intertwining between spin and charge sectors is partly weakened at the Kondo temperature due to a progressive reduction of the effective spin-fermion coupling of local magnetic fluctuations in the low frequency regime. Our analysis, thus, clarifies the precise mechanism through which the physical information is transferred between different scattering channels of interacting electron problems and highlights the pivotal role played by such an intertwining in the physics of correlated electrons beyond the perturbative regime.



# 1  Introduction

Feynman diagrammatics is often the formalism of choice to describe the striking physical properties of correlated quantum materials, since its quantum-field-theoretical framework is naturally suited for treating the enormously large number of degrees of freedom of the many-electron problem. However, formulating such diagrammatics in the context of condensed matter physics suffers from the absence of an *a priori* small expansion parameter. This drawback has direct consequences in limiting the range of applicability of perturbative approaches [1–3] for correlated electron systems. More specifically, the breakdown of the (self-consistent) perturbative expansion occurring for increasing values of repulsive interactions is formally reflected in the intrinsic multivaluedness of the Luttinger-Ward functional [2–13] and, at the same time, in multiple divergences of two-particle irreducible vertex functions in the charge and the pairing scattering channels [1,3,7,9,10,14–23].

Beyond these formal aspects and their relevant algorithmic consequences, the physical mechanisms underlying the breakdown of the perturbation expansions have been the central object of several studies [3,16,20,21,23–28] in the most recent years. Particularly relevant for this work is the (essentially heuristic) observation [3,25,28] of a link between the breakdown of perturbation theory and the formation of a local magnetic moment or, to be more precise, its strong *intertwining* with local charge (or pairing) fluctuations [3,25]. In Ref. [25] the authors have identified characteristic structures in the Matsubara frequency space of the generalized charge susceptibility, which they related to the formation and the subsequent screening of a local magnetic moment.

Within this work, our goal is to demonstrate the validity of this intuitive observation and, most importantly, to precisely explain the microscopic mechanisms driving the non-perturbative physics in the charge sector. To this aim, we will exploit the single-boson exchange (SBE) formalism [29,30] to decompose and analyze the generalized charge susceptibilities of several fundamental models of interest, ranging from the most elementary, analytically solvable Hubbard atom (HA) to the numerically exactly solvable Anderson impurity model (AIM) and the Hubbard model (HM) on the Bethe lattice solved within dynamical mean-field theory (DMFT) [31].

Different from more standard decomposition schemes mainly used in the past, such as those based on the parquet equations [16], the SBE decomposition is applicable in the *whole* parameter regime, from weak to strong-coupling, see, e.g., Refs. [27,32–35], providing hence an ideal framework for this study. Furthermore, the SBE formalism is physically transparent, allowing for a particularly insightful identification of the microscopic processes at play, similar to the framework of Ref. [35].

The paper is organized as follows: In Sec. 2, we provide the definition of the two-particle quantities of interests for our work, a short presentation of the models studied and of the numerical methods used to solve them, and a concise introduction of the SBE formalism adopted in our analysis. Afterwards, in Sec. 3, we present our numerical results for the generalized charge susceptibility as well as the corresponding SBE decomposition. Our analysis ranges from the perturbative high-temperature regime, to the local-moment regime at intermediate temperatures down to the low-temperature Kondo regime, similarly as in Ref. [25]. Eventually, in Sec. 4, we summarize our main findings and outline, as an outlook, interesting research questions to be addressed in the future.

## 2 Formalism

### 2.1 Formal definitions

The central quantity, which will be computed and systematically analyzed in this work, is the generalized local/on-site charge susceptibility $\tilde{\chi}^{\text{ch}}_{\nu\nu',\omega=0}$ as a matrix in the fermionic Matsubara frequency space $[\nu_n \equiv \nu = (2n-1)\pi T, \nu'_{n'} \equiv \nu' = (2n'-1)\pi T]$, with $n, n' \in \mathbb{Z}$ and $T$ the temperature of the systems considered (see below), for a zero value[1] of the (bosonic) transfer frequency $\omega$.

The definition of the generalized susceptibility in terms of one-particle $G_\nu$ and two-particle Green's functions $G^{(2)}_{\nu\nu',\omega}$ reads as follows [38,39]:

$$\tilde{\chi}^{\text{ch}}_{\nu\nu',\omega=0} = -2\beta G_\nu G_{\nu'}\delta_{\omega,0} + \left(G^{(2)}_{\nu\nu'\omega=0,\uparrow\uparrow} + G^{(2)}_{\nu\nu'\omega=0,\uparrow\downarrow}\right), \tag{1}$$

where $G^{(2)}_{\nu\nu'\omega,\sigma\sigma'}$ is defined as [38–40]

$$G^{(2)}_{\nu\nu'\omega,\sigma\sigma'} = \int_0^\beta d\tau_1 d\tau_2 d\tau_3 e^{i\nu\tau_1} e^{-i(\nu+\omega)\tau_2} e^{i(\nu'+\omega)\tau_3} \langle \mathcal{T}_\tau \hat{c}_\sigma(\tau_1)\hat{c}^\dagger_\sigma(\tau_2)\hat{c}_{\sigma'}(\tau_3)\hat{c}^\dagger_{\sigma'}(0)\rangle, \tag{2}$$

where $\langle\cdots\rangle = Tr(e^{-\beta(\mathcal{H}-\mu N)}\cdots/Z)$ is the thermal average ($\beta$ denoting the inverse temperature $\frac{1}{T}$, $\mathcal{H}$ the Hamiltonian operator and $N$ the total particle number operator), $Z = Tr(e^{-\beta(\mathcal{H}-\mu N)})$ the partition function, $\sigma =\uparrow,\downarrow$ the spin orientation along the quantization axis, $\mathcal{T}_\tau$ the imaginary time-ordering operator and $G_\nu$ reads as [38–40]

$$G_\nu \equiv G_{\nu,\sigma=\uparrow} = -\int_0^\beta d\tau e^{i\nu\tau}\langle \mathcal{T}_\tau \hat{c}_\sigma(\tau)\hat{c}^\dagger_\sigma(0)\rangle, \tag{3}$$

where $G_{\nu,\sigma=\uparrow} = G_{\nu,\sigma=\downarrow}$ due to the SU(2) symmetry [38]. For the purpose of our study it is important to recall that the physical response of the system (the physical susceptibility $\chi_\omega$) is

---

[1]The reason of this specific choice is twofold: (i) the $\omega = 0$ is directly linked to the behavior of the static/isothermal response function of the systems, which allows for particularly easy physical interpretation and for a direct readability [36,37] of the conserved quantities in the system considered; (ii) the breakdown of the self-consistent many-electron perturbation expansions [1–3] manifests itself first in the scattering amplitude associated with a vanishing transfer frequency [3,7,10].

related to the associated generalized one through the following relation, reported here explicitly for the charge sector [31, 39]:

$$\chi_\omega^{\mathrm{ch}} \equiv \int_0^\beta d\tau\, e^{-i\omega\tau} \langle [\hat{n}(\tau) - \langle \hat{n} \rangle][\hat{n}(0) - \langle \hat{n} \rangle] \rangle = \frac{2}{\beta^2} \sum_{\nu,\nu'} \tilde{\chi}_{\nu\nu',\omega}^{\mathrm{ch}}, \tag{4}$$

where $\hat{n}(\tau) = \sum_{\sigma=\uparrow,\downarrow} \hat{c}_\sigma^\dagger(\tau)\hat{c}_\sigma(\tau)$ represents the density operator expressed in terms of the corresponding creation/annihilation operators of an electron with spin-orientation $\sigma$, and $\langle n \rangle$ its grand-canonical expectation value.

In the course of this work, we will compute and analyze the multifaceted behavior of $\tilde{\chi}_{\nu\nu',\omega=0}^{\mathrm{ch}}$ in fundamental models for correlated electrons (i.e., the HA, the AIM and a DMFT solution of the HM on the Bethe lattice) in three relevant regimes: (i) the perturbative regime at high temperatures, (ii) the local-moment regime at intermediate temperatures and –where it applies– the Kondo regime at low temperatures.

## 2.2 Models

The simplest, yet insightful, model considered is the Hubbard atom (HA), which consists of a single interacting site without any bath connected to it. Its Hamiltonian reads:

$$\hat{H}^{\mathrm{HA}} = U\, \hat{n}_\uparrow \hat{n}_\downarrow, \tag{5}$$

where $\hat{n}_\sigma = \hat{n}_\sigma(\tau = 0)$ are the occupation operators and $U$ is the interaction strength. Due to its simplicity its correlation functions can be computed analytically [10, 41].

The second model considered in order of complexity is the Anderson impurity model (AIM), which consists of a single interacting site hybridized with a non-interacting electronic bath [42],

$$\hat{H}^{\mathrm{AIM}} = \sum_{k\sigma} \epsilon_k\, \hat{a}_{k\sigma}^\dagger \hat{a}_{k\sigma} + \sum_{k\sigma} V_k\, (\hat{c}_\sigma^\dagger \hat{a}_{k\sigma} + \hat{a}_{k\sigma}^\dagger \hat{c}_\sigma) + U\, \hat{n}_\uparrow \hat{n}_\downarrow, \tag{6}$$

where $\hat{a}_{k\sigma}^\dagger, \hat{a}_{k\sigma}$ are the creation and annihilation operator of the bath states with energy $\epsilon_k$ and spin $\sigma$, $\hat{c}_\sigma^\dagger, \hat{c}_\sigma$ are the creation and annihilation operator of charge carriers on the interacting impurity site, and $V_k$ are the hybridization parameters determining the hopping from the bath states to the interacting site. For our study we adopt the following parameters: a non-interacting bath with a flat density of states of half-bandwidth 10 (in our unit of energy) and an energy-independent hybridization parameter $V_k = V = 2$. For the values of $U$ considered, this corresponds [17, 25, 42, 43] to an AIM in the wide-band limit, which features all relevant regimes we are interested in. Starting with the high-$T$ perturbative one, to the local moment regime at intermediate temperature and a Kondo screening behavior at low $T$.

As a third model, we consider the DMFT [31] solution of the Hubbard model (HM) on a Bethe lattice with infinite coordination number, whose Hamiltonian reads:

$$\hat{H}^{\mathrm{Hubbard}} = \sum_{\langle i,j \rangle} t_{ij} \hat{c}_i^\dagger \hat{c}_j + U \sum_i \hat{n}_{i\uparrow} \hat{n}_{i\downarrow}, \tag{7}$$

where $t_{ij}$ denotes the hopping amplitude between neighboring sites $i, j$ and $\hat{n}_{i,\sigma}$ the occupation operator at site $i$. At the value of the $U$ we consider in this work ($U = 2.2$, expressed in unit of the Bethe-halfbandwidth $2t = 1$), its DMFT solution displays a low-temperature crossover between a coherent Fermi liquid and a incoherent/bad-metallic phase [31].

In all these models, we fix $\mu = \frac{U}{2}$, enforcing the half-filling condition, where correlation effects are typically the strongest. For the HM we consider the half-bandwidth as our unit of energy. From an algorithmic point of view, as an impurity solver for the AIM, as well as for the auxiliary AIM of the DMFT solution, we use the continuous-time quantum Monte Carlo (CT-QMC) [44] solver of w2dynamics [45, 46].

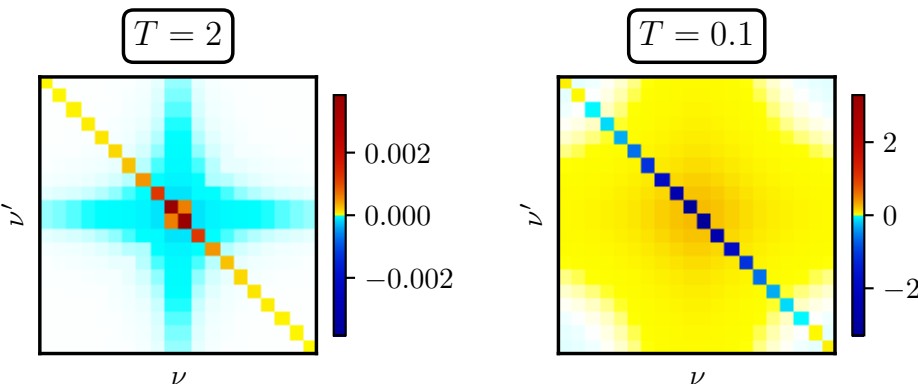

Figure 1: Representative examples of the main frequency structures displayed by the generalized charge susceptibility $\tilde{\chi}^{\text{ch}}_{\nu\nu'}$ of the HA at $U = 5.75$ in its perturbative/ high-$T$ regime (left panel) and its non-perturbative/local moment low-T regime (right panel). The perturbative result is characterized by strictly positive diagonal entries and by a faint, mostly negative off-diagonal background. Remarkably the main low-frequency structures of $\tilde{\chi}^{\text{ch}}_{\nu\nu'}$ appear to be completely reversed in the non-perturbative/low-T regime: We observe negative diagonal entries and a faint positive off-diagonal background.

## 2.3 Single-boson exchange decomposition of the generalized susceptibility

The generalized charge susceptibility represents, due to the high sensitivity of its frequency structure to changes in the physical properties of the system, a reliable "bellwether" for the onset of the non-perturbative regime [25]. This significant aspect is well exemplified by the local generalized charge susceptibility computed in two paradigmatic limiting (perturbative/non-perturbative) cases, which are shown in Fig. 1. Here, the generalized susceptibility $\tilde{\chi}^{\text{ch}}_{\nu\nu'}$ for zero transfer frequency is plotted as a matrix[2] for the HA in the perturbative/high-$T$ (left panel) and the non-perturbative/low-$T$ (right panel) region. Well visible in the perturbative regime is the positive (yellow to red) diagonal emerging on top of a faint negative (turquoise to blue) background. In comparison, in the non-perturbative regime, the structure appears reversed: Now the faint background is purely positive while the diagonal at low frequencies becomes negative. The strong suppression of the diagonal entries in $\tilde{\chi}^{\text{ch}}_{\nu\nu'}$, even down to negative values, drives the breakdown of the self-consistent perturbative description [3, 16, 17]. In fact, it is responsible for several sign flips (from positive to negative) of the eigenvalues of the generalized susceptibility and, hence, for corresponding divergences of two-particle irreducible vertex functions in the corresponding channel [1]. Evidently, such divergences pose considerable problems to all diagrammatic many-electron approximations which rely on the extraction and/or the manipulation of two-particle irreducible vertices in the nonperturbative regime. Among these, we should certainly recall the diagrammatic approaches based on parquet equations [47, 48], such as the dynamical vertex approximation [49–51] and the QUADRILEX scheme [52], though closely related difficulties can also affect self-consistent diagrammatic Monte Carlo and nested cluster algorithms [2, 9]. On a more physical point of view, the qualitative change in the frequency structure of $\tilde{\chi}^{\text{ch}}_{\nu\nu'}$ has been interpreted as a "fingerprint" [25] of the local moment formation,[3] which gradually takes place by increasing $U$

---

[2]Here we adopt the convention in which the diagonal runs from the upper left corner to the lower right corner.

[3]While the intrinsic crossover nature of the local moment formation (as well as of its screening) prevents the definition of a unique criterion for delimiting its borders, the $U$ and $T$ values at which the lowest-frequency diagonal entries of $\tilde{\chi}^{\text{ch}}_{\nu\nu'}$ are suppressed below a certain threshold (e.g., where they become negative) outline a region in

at fixed temperature for all the model considered.[4] This interpretation, also consistent with specific observations made in [3, 16, 17], was essentially of empirical nature, i.e., it relies on the identification of common trends in different sets of numerical calculations. Evidently, this leaves open the fundamental question of *why* the suppression of on-site charge fluctuations occurs in the specific way it is observed.

The importance of rigorously clarifying this issue is twofold: First, it is necessary to render the abovementioned interpretation generally applicable to a broad class of physical situations, i.e. beyond specific sets of numerical data. Second, it allows to elucidate the precise mechanism through which the scattering processes underlying the local moment formation are responsible for the breakdown of the many-electron perturbation theory. Indeed, the theoretical framework would completely change, if the suppression effects associated to the local moment formation were encoded differently in the generalized charge susceptibility $\tilde{\chi}^{\mathrm{ch}}_{\nu\nu'}$, e.g. via an uniform suppression of all its matrix elements. In this case, a consistent description of the local moment physics could have been indeed achieved by means of purely perturbative approaches. At the same time, relevant physical phenomena such as the phase-separation instabilities occurring in the proximity of the Mott metal-insulator transition described in DMFT [55, 56], which have been directly ascribed to the appearance of *negative* eigenvalues in the generalized on-site charge susceptibility [21, 24, 57, 58], could no longer take place.

The aim of this paper is then to rigorously explain the physical roots of the frequency structures emerging in the generalized charge susceptibility in different parameter regimes. In order to do so, we start by recalling that the generalized susceptibilities of many-electron systems can always be decomposed into terms arising from the independent propagation of a particle and of a hole, usually referred as (*"bubble"*), and into a term encoding all scattering processes between the particle and the hole (*"vertex correction"*) [38]. In the specific case of the local static generalized charge susceptibility, this relation reads:

$$\tilde{\chi}^{\mathrm{ch}}_{\nu\nu',\omega=0} = -\beta[G_\nu]^2 \delta_{\nu\nu'} - [G_\nu]^2 F^{\mathrm{ch}}_{\nu\nu',\omega=0}[G_{\nu'}]^2, \tag{8}$$

where the bubble term is the first contribution on the r.h.s. of the equation and $F^{\mathrm{ch}}_{\nu\nu',\omega=0}$ denotes the full scattering amplitude in the charge scattering channel, i.e. the full two-particle charge vertex. It is useful to emphasize here that while the two-particle propagation described by the "bubble" term of Eq. (8) corresponds to incoming/outgoing electrons/holes with the same energy ($\nu = \nu'$), the scattering processes encoded in the vertex correction term do not only change the value of the diagonal entries of $\tilde{\chi}^{\mathrm{ch}}_{\nu\nu',\omega=0}$ but are also responsible for the emergence of finite off-diagonal elements ($\nu \neq \nu'$).

In order to perform a rigorous analysis and access the fundamental mechanisms at work in the different perturbative/non-perturbative physical regimes we will exploit the Single-Boson Exchange (SBE) formalism. In this way, we can decompose the numerical results for $\tilde{\chi}^{\mathrm{ch}}_{\nu\nu',\omega=0}$, and in particular its vertex-correction part, in terms of processes involving the exchange of *single well-defined* bosonic modes. In practice, this means to apply the following decomposition [29] to the full scattering amplitude (or full vertex function) $F^{\mathrm{ch}}$ appearing on the r.h.s of

---

the corresponding phase-diagrams, whose shape is qualitatively consistent with the crossover borders of the local moment regime defined through different criteria in the recent literature [28, 53], see also [54] as well as the very last part of the discussion in Appendix E.

[4]With the obvious exception of the first order jump at the Mott transition in the DMFT solution of the HM, which unavoidably affects [23] also the behavior of two-particle quantities, but which is not specifically relevant for our analysis.

Eq. (8), i.e., explicitly: $F^{\text{ch}}_{\nu\nu'\omega}$ reads:

$$
\begin{aligned}
F^{\text{ch}}_{\nu\nu'\omega} = {} & \phi^{\text{Uirr, ch}}_{\nu\nu'\omega} + \lambda^{\text{ch}}_{\nu\omega} w^{\text{ch}}_{\omega} \lambda^{\text{ch}}_{\nu'\omega} \\
& - \frac{1}{2}\big(\lambda^{\text{ch}}_{\nu,\nu'-\nu} w^{\text{ch}}_{\nu'-\nu} \lambda^{\text{ch}}_{\nu+\omega,\nu'-\nu} + 3\lambda^{\text{sp}}_{\nu,\nu'-\nu} w^{\text{sp}}_{\nu'-\nu} \lambda^{\text{sp}}_{\nu+\omega,\nu'-\nu}\big) \\
& + \frac{1}{2}\lambda^{\text{si}}_{\nu,\nu+\nu'+\omega} w^{\text{si}}_{\nu+\nu'+\omega} \lambda^{\text{si}}_{\nu',\nu+\nu'+\omega} - 2U,
\end{aligned}
\tag{9}
$$

where

$$
w^{\alpha}_{\omega} = U^{\alpha} - \frac{1}{2}U^{\alpha}\chi^{\alpha}_{\omega}U^{\alpha}
\tag{10}
$$

(depicted with a wiggly line in our diagrammatic representation), describes a collective bosonic excitation of different kind ($\alpha = $ ch,sp,si, with sp denoting the spin channel and si the singlet pairing channel), $U^{\text{ch}} = U$, $U^{\text{sp}} = -U$ and $U^{\text{si}} = 2U$, and $\lambda^{\alpha}_{\nu\omega}$ are the Hedin vertices coupling the external legs to the exchange boson (see appendix A for definitions) [59]. Eventually, $\phi^{\text{Uirr, ch}}_{\nu\nu'\omega}$ encodes all scattering processes associated with the exchange of *more than one* bosonic-mode (multi-boson processes in the charge channel). This decomposition has been explicitly derived in [29] and already been exploited in different contexts [27, 33, 34, 60–62]. It formally corresponds to a classification of the two-particle diagrams of the full scattering amplitude ($F^{\text{ch}}$) in terms of their irreducibility w.r.t. a cut of an $U$-interaction line.

For the scope of this paper, we will use Eq. (9) to explicitly recast Eq. (8) as

$$
\begin{aligned}
\tilde{\chi}^{\text{ch}}_{\nu\nu',\omega=0} = {} & -\beta[G_{\nu}]^2\delta_{\nu\nu'} && \text{bubble} \\
& -[G_{\nu}]^2\phi^{\text{Uirr, ch}}_{\nu\nu',\omega=0}[G_{\nu'}]^2 && \text{U-irr} \\
& -[G_{\nu}]^2\lambda^{\text{ch}}_{\nu,\omega=0}w^{\text{ch}}_{\omega=0}\lambda^{\text{ch}}_{\nu',\omega=0}[G_{\nu'}]^2 && \text{charge 1} \\
& +\frac{1}{2}[G_{\nu}]^2\lambda^{\text{ch}}_{\nu,\nu'-\nu}w^{\text{ch}}_{\nu'-\nu}\lambda^{\text{ch}}_{\nu,\nu'-\nu}[G_{\nu'}]^2 && \text{charge 2} \\
& +\frac{3}{2}[G_{\nu}]^2\lambda^{\text{sp}}_{\nu,\nu'-\nu}w^{\text{sp}}_{\nu'-\nu}\lambda^{\text{sp}}_{\nu,\nu'-\nu}[G_{\nu'}]^2 && \text{spin} \\
& -\frac{1}{2}[G_{\nu}]^2\lambda^{\text{si}}_{\nu,\nu+\nu'}w^{\text{si}}_{\nu+\nu'}\lambda^{\text{si}}_{\nu',\nu+\nu'}[G_{\nu'}]^2 && \text{singlet} \\
& +2U[G_{\nu}]^2[G_{\nu'}]^2 && \text{double counting (DC)},
\end{aligned}
\tag{11}
$$

which allows for a transparent identification of all contributions[5] to $\tilde{\chi}^{\text{ch}}_{\nu\nu',\omega=0}$ in terms of the bubble contribution, the exchanges of a single collective/bosonic excitation ($\alpha = $ ch, sp, si) and of more complicated multi-boson processes.

To illustrate how this decomposition of $\tilde{\chi}^{\text{ch}}_{\nu\nu'}$ works in practice, we apply Eq. (11) to the perturbative/high-T data set of the HA, shown in the left panel of Fig. 1. As explicitly shown by the results reported in Fig. 2, all the different contributions to $\tilde{\chi}^{\text{ch}}_{\nu\nu'}$ appearing in Eq. (11) have been computed separately (with the exception of the multiboson term, which is simply evaluated via the difference of the sum of all terms w.r.t. to $\tilde{\chi}^{\text{ch}}_{\nu\nu'}$) and have been identified by the related Feynman-diagrammatic expression in the corresponding panels. In this way, the specific role in shaping the final structure of $\tilde{\chi}^{\text{ch}}_{\nu\nu'}$ played by the different SBE as well as by the multiboson scattering processes (reported on the leftmost panel of the first row) can be quantitatively and unambiguously determined.

It is worth noticing that, at half-filling, different SBE contributions have definite signs, as it can be analytically understood from Eqs. (10) and (11). For instance, the spin contribution displays an overall negative sign, due to $w^{\text{sp}}_{\nu'-\nu} < 0$, consistent with the iterated perturbation theory analysis for the full vertex of Ref. [26]. Evidently, for our work, beyond the

---

[5]With the formal exception of the double-counting (DC) term, i.e. of the last term in Eq. (11), which yields only a marginal contribution in the most insightful intermediate-to-strong coupling regime.

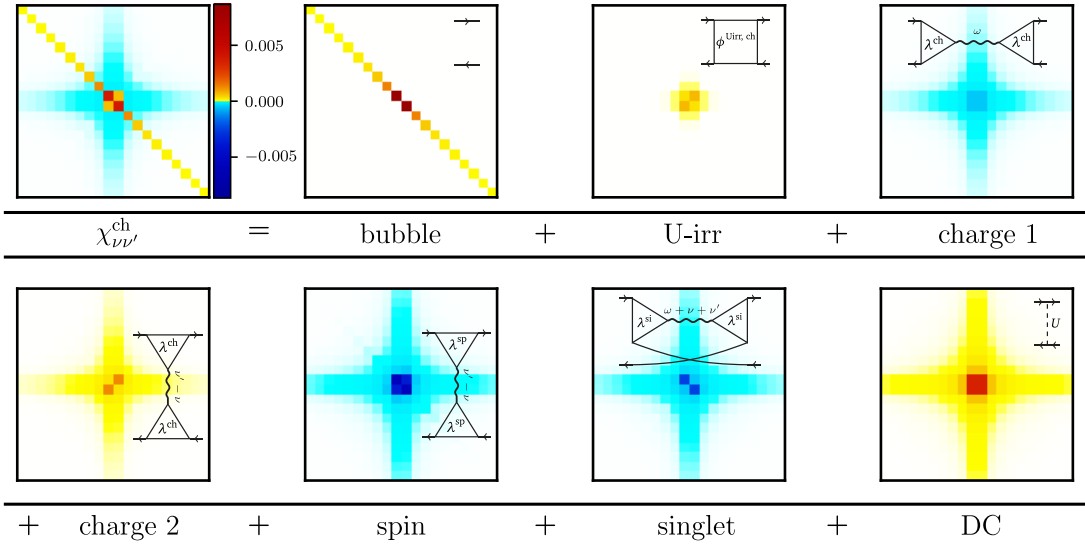

Figure 2: Example of the SBE decomposition according to Eq. (11), applied to the $\tilde{\chi}_{\nu\nu'}^{\text{ch}}$ data for the HA at $T = 2$ and $U = 5.75$, shown in the left panel of Fig. 1. The full $\tilde{\chi}_{\nu\nu'}^{\text{ch}}$ is reported here in the top left panel, followed by its bubble term, its multiboson/U-irreducible contribution, its different SBE terms (charge, spin, singlet) and, eventually, by the double counting contribution (schematic Feynman diagrams of the corresponding processes are depicted in the top-right corner of all panels).

overall/definite signs of the different terms, their relative magnitudes play a crucial role. In this respect, we mention already here that we will mostly focus on the bubble, the fully U-irreducible, and the spin contribution of the SBE decomposition, since those three terms will turn out to be dominant in the intermediate-to-strong coupling (non-perturbative) parameter regimes.

## 3 Results

In this section we study the mechanisms controlling the frequency structures of the generalized charge susceptibility $\tilde{\chi}_{\nu\nu'}^{\text{ch}}$ in different physical regimes. In particular, by exploiting its SBE decomposition defined in Eq. (11) in terms of scattering processes corresponding to the exchange of zero, single, and multiple bosonic collective modes, we identify the role played by the different microscopic mechanisms and by their intertwining. In this context, we can unambiguously link the emergence of characteristic frequency structures [25] of $\tilde{\chi}_{\nu\nu'}^{\text{ch}}$ of purely non-perturbative nature (e.g., the strongly negative diagonal as in the right panel Fig. 1) to specific physical properties (and constraints) of the strong-coupling regime.

In the following, we proceed step by step to illustrate our results throughout three different temperature regimes: First, the high-temperature regime, which is typically well described by perturbative approaches, second, the intermediate temperature regime, where a local moment is formed in all the models considered (HA, AIM and HM solved by DMFT), and third, the low-temperature regime, where the local moment of the AIM and HM is gradually screened. Evidently, the physics we consider here is not characterized by any abrupt phase-transition, but rather by crossovers between different regimes. The borders of such crossovers depend, to a certain extent, on the different criteria adopted, and in this respect, we refer the readers, beyond the standard references such as [42, 63] to the more recent analyses [25, 28, 35, 54]

on this subject. Instead, for our scopes it is important to choose parameter sets, where one can safely assume to be in one of the different regimes. For instance, as electronic localization effects will become only predominant for $T \ll \frac{U}{2}$ we choose for all three models considered $T \sim O(\frac{U}{2})$, in order to study the high-$T$/perturbative regime. Specifically, in the units chosen for this work $T = 2$ for HA and AIM, and $T = \frac{2}{3}$ for the HM solved with DMFT.[6] As for the local moment regime, we choose $T = 0.1$, which satisfies the following condition $T_K \ll T \ll \frac{U}{2}$, where $T_K$ represents the corresponding Kondo temperature of the problem considered (namely, $T_K = 0$ for the HA, $T_K \simeq 0.015$ for our AIM [17,25], and $T_K \sim T_{\text{coh}} \simeq 0.02$ for the DMFT solution of the HM considered [25,43]). Finally, in the two cases, AIM and HM solved by DMFT, where low-$T$ screening processes do occur, we have chosen representative temperatures (respectively $T = \frac{1}{60} \simeq 0.0167$ and $T = 0.02$) very close to the corresponding $T_K$ (or "effective" $T_K$ in DMFT, which yields the coherent scale, $T_{\text{coh}}$, of the underlying Fermi-liquid state). The location of these selected temperature-values for each model is compared with the underlying physics in Fig. 14, s. Appendix E, where the explicit $T$-dependence of the physical (static) on-site spin and charge susceptibilities are reported on the whole temperature range of the three models.

## 3.1 Perturbative/high-temperature regime

We begin by analyzing the high-temperature regimes of the three cases considered (HA, AIM, DMFT solution of the HM), where the temperature has been fixed to values larger and/or comparable to the other relevant energy scales of the systems (s. caption of Fig. 3 for details). In this context [25,38,64,65], all frequency-dependent correlation functions of the systems, including $\tilde{\chi}^{\text{ch}}_{\nu\nu'}$ are qualitatively well described by perturbative approaches like the RPA or the parquet approximation (PA) [47] (not shown).

The results of our calculations of $\tilde{\chi}^{\text{ch}}_{\nu\nu'}$, as well of its SBE decomposition according to Eq. (11), are reported in Fig. 3 and Fig. 4. In particular, in Fig. 3 the data of the diagonal entries $\tilde{\chi}^{\text{ch}}_{\nu,\nu'=\nu}$ of the generalized susceptibility are plotted (black squares) as a function of $\nu$ and analyzed via their SBE decomposition. The different contributions to the latter are displayed as symbols with different colors.

We immediately notice that the diagonal entries are positive in all three models considered, consistent with $\tilde{\chi}^{\text{ch}}_{\nu\nu'}$ being a positive (and almost diagonal matrix) in the weak-coupling limit. Furthermore, as expected in a perturbative regime, the corresponding SBE decomposition looks qualitatively very similar in the different models: Its major (positive) contribution is always represented by the bubble term[7] $\propto -\beta G_\nu^2 > 0$ (gray empty circles), while several SBE and double-counting terms (dots of different colors), which partly compensate each other, yield an overall negative (suppressing) contribution. As for the latter, we notice that the major damping effects of $\tilde{\chi}^{\text{ch}}_{\nu,\nu'=\nu}$ originate from the SBE processes in the spin channel (red dots/solid lines), followed by the corresponding ones in the singlet pairing channel (green dots). As for the predominant spin term, we disclose here that, in the following, it will be very insightful to separately consider the contribution directly proportional to the physical magnetic suscep-tibility, i.e. $\propto U^2 \chi^{\text{sp}}_\omega$ (which arises from inserting Eq. 10 into the spin term of Eq. 11, i.e. $\frac{3}{2}[G_\nu]^2 \lambda^{\text{sp}}_{\nu,\nu'-\nu} U^2 \chi^{\text{sp}}_{\nu'-\nu} \lambda^{\text{sp}}_{\nu,\nu'-\nu}[G_{\nu'}]^2)$. This specific contribution to the spin SBE is shown as red dotted line in Fig. 3 as well as in all the corresponding figures throughout this work.

We now extend our analysis to the full frequency dependence in $\nu$ and $\nu'$, shown in Fig. 4 (for the static $\omega = 0$ case). In particular, in the first column, $\tilde{\chi}^{\text{ch}}_{\nu\nu'}$ for the three models considered is reported in the significant frequency range, demonstrating how, in this parameter

---

[6]While even higher temperatures could be chosen as representative for this regime, this choice represents a good compromise. It allows us not to loose too much information in Matsubara frequency space, as the first Matsubara frequency scales as $\pi T$.

[7]Note that the large values of $G_\nu^2$ obtained in the (insulating) HA are simply a consequence of the high temperature limit considered, i.e. $\nu_0 = \pi T \approx U$.

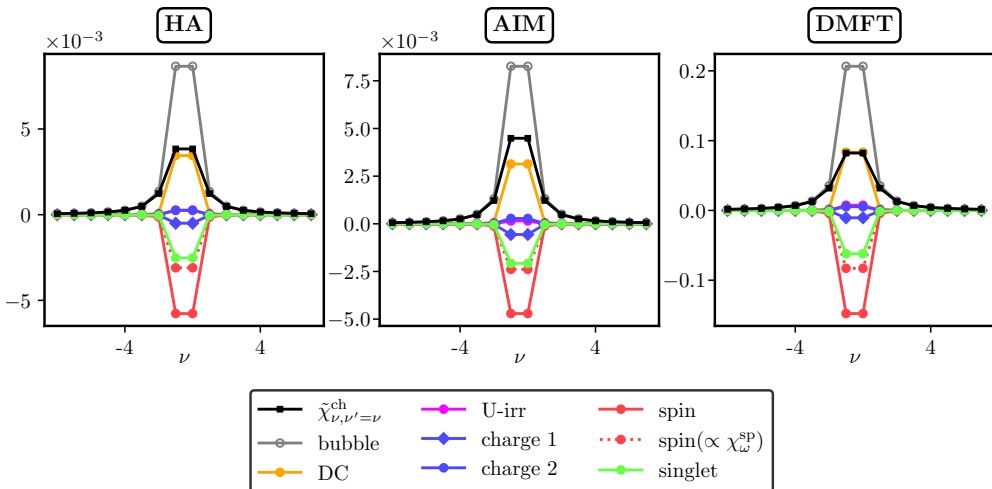

Figure 3: Diagonal elements of $\tilde{\chi}^{\mathrm{ch}}_{\nu\nu'}$ (black squares) as a function of the Matsubara frequency $\nu$ and their corresponding SBE-decomposition (symbols in different colors), according to Eq. (11) (cf. also Fig. 2) in the perturbative/high-$T$ regime ($T = 2, U = 5.75$ for HA and AIM and $T = 2/3, U = 2.2$ for the DMFT solution). The red dotted line shows the spin-contribution directly proportional to $\chi^{\mathrm{sp}}_{\omega}$, i.e. $\frac{3}{2}[G_\nu]^2\lambda^{\mathrm{sp}}_{\nu,\nu'-\nu}U^2\chi^{\mathrm{sp}}_{\nu'-\nu}\lambda^{\mathrm{sp}}_{\nu,\nu'-\nu}[G_{\nu'}]^2$. All three models show a similar behavior, where no specific term of Eq. (11) is predominating: The large positive behavior of the bubble term is reduced by the overall effects of the vertex-correction terms, whereas the most negative contributions arise from processes associated to the exchange of a spin or a singlet-pairing collective excitation.

regime, the above-mentioned similarity of the results holds in the whole frequency space. Consistently, the full SBE decomposition explicitly shown in Fig. 2 can be *de facto* applied to all three data sets presented in Fig. 4. Such decomposition evidently confirms the considerations made above: a generalized charge susceptibility dominated by a positive diagonal bubble term on top of a faint negative background, where the latter results from the partial compensations of different SBE, and double-counting terms.

While a quantitative decomposition of $\tilde{\chi}^{\mathrm{ch}}_{\nu\nu'}$ in terms of Eq. (11) can be systematically performed whenever needed as in Fig. 2, it is convenient here to restrict our focus to the physically most significant contributions, in perspective of our following study of the more challenging non-perturbative regimes. This is done in the second-to-fourth column of panels in Fig. 4, where, for each model, we report the (positive) contributions arising from the bubble term (second column), from the multiboson processes, which are only sizable at the lowest frequencies (third column), as well as the major damping (i.e., negative) contribution to the local charge fluctuations arising from SBE processes.

By considering Eqs. 10 and 11 and neglecting the frequency dependence of the Hedin vertices, one observes that the terms linear in $U$ from the charge, the spin and the DC term cancel each other. The numerical data in the high-$T$ regime considered here appears, indeed, consistent with this asymptotic analytical property. Overall, this renders the remaining spin term, which is directly proportional to $U^2\chi^{\mathrm{sp}}_{\omega}$ together with the singlet term the dominating negative contributions. The sum of these two terms is reported in the fourth column of Fig. 4. The plots in the second-to-fourth column panel allow thus to identify in a glimpse the predominant diagrams shaping the generalized susceptibility at weak-coupling. In the following subsections we are going to apply our strategy to the analysis of the more interesting strong-coupling/non-perturbative regimes.

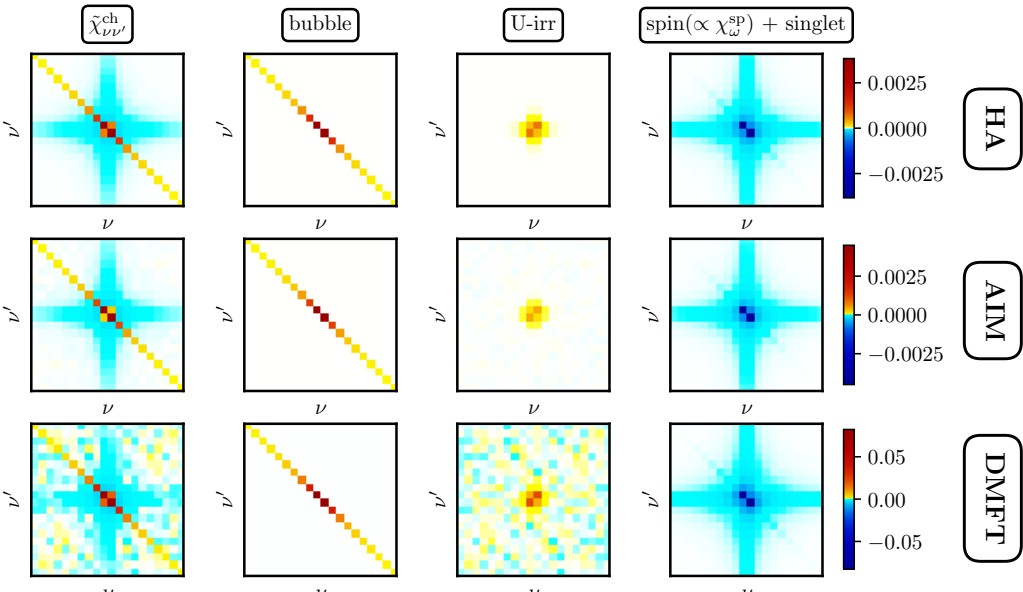

Figure 4: Results for our SBE decomposition of $\tilde{\chi}^{\text{ch}}_{\nu\nu'}$ in the HA (top row), the AIM (center row) and the DMFT solution of the HM (bottom row) in the perturbative regime (same parameters as Fig. 3). The left panels report the generalized charge susceptibility for the first 12 Matsubara frequencies in $\nu$ and $\nu'$, which is similar for all three models in this regime. The second column of panels from the left show the (strictly diagonal and positive) bubble contribution. The third column of panels shows the U-irr/multiboson contribution, which, in the perturbative regime, appears negligible everywhere except at the lowest frequencies. The last column shows the spin contribution directly connected to $\chi^{\text{sp}}_{\omega}$ plus the singlet contribution, see text and Eq. (11).

## 3.2 Intermediate-temperature/local moment regime

As the next step, we focus on intermediate temperatures, for which, in the three cases considered, local magnetic moments are formed as a result of the on-site repulsion $U$. Following the same steps as in the previous subsection, we start our analysis of $\tilde{\chi}^{\text{ch}}_{\nu\nu'}$ by considering its diagonal elements ($\nu' = \nu$). These are shown in Fig. 5 together with their complete SBE decomposition of $\tilde{\chi}^{\text{ch}}_{\nu\nu'}$, similarly as in Fig. 3. We can immediately observe that, for all three models, the diagonal entries of $\tilde{\chi}^{\text{ch}}_{\nu\nu'}$ (black dots) become negative in the low-frequency regime, while turning back to positive values at intermediate/large frequencies [25].

This qualitative change w.r.t. the high-$T$/perturbative case, as shown in Fig. 3, is important for a twofold reason. First, from a physical point of view, the negative diagonal elements play a major role in the suppression of the on-site static charge response which characterizes this regime, when the corresponding sum over all fermionic Matsubara frequencies ($\nu, \nu'$) is taken in Eq. (4) [25]. Second, from a more formal perspective, the suppression of the diagonal entries of $\tilde{\chi}^{\text{ch}}_{\nu\nu'}$ down to negative values drives the zero crossing, and the subsequent negativity, of some of its eigenvalues [25]. This has direct consequences for calculating the two-particle irreducible vertex in the charge channel from the inversion of the corresponding Bethe-Salpeter equation: $\Gamma^{\text{ch}}_{\nu\nu'\omega} = \beta^2([\tilde{\chi}^{\text{ch}}_{\omega}]^{-1}_{\nu\nu'} - [\tilde{\chi}^{\text{ch},0}_{\omega}]^{-1}_{\nu\nu'})$. In fact, zero-crossing eigenvalues of $\tilde{\chi}^{\text{ch}}_{\nu\nu'}$ render the Bethe-Salpeter equation non-invertible, triggering (multiple) divergences of the two-particle irreducible vertex in the charge channel [1, 3, 7, 9, 10, 14–17, 19, 20] and, hence, the associated breakdown of the self-consistent perturbation theory [3].

As mentioned in Sec. 2, it has been recently proposed [25] to consider the appearance of negative diagonal elements in $\tilde{\chi}^{ch}_{\nu\nu'}$ as a characteristic "fingerprint" of the local moment formation in the charge channel. However, a microscopic understanding of this intuitive characterization has been lacking hitherto. For instance, it was unclear why the suppression of the on-site charge fluctuations should necessarily occur via a specific, selective suppression of the diagonal elements of the corresponding generalized susceptibility, when it is physically associated to the local moment formation. In principle, the same reduction of the local on-site charge response could be formally achieved in several other plausible ways (e.g., by an overall reduction of all matrix elements of $\tilde{\chi}^{ch}_{\nu\nu'}$). Our analysis below finally clarifies, among other points, these relevant questions.

Proceeding by examining our SBE decomposition of the results (data points of different colors in Fig. 5), we note the prominence of two contributions: (i) a major negative term associated to the SBE in the spin term (red dots) and (ii) a smaller positive bubble term (gray open circles). More specifically, we observe that the spin-exchange damping in the local-moment regime is *de facto* coinciding, in the three models, with the part proportional to the spin-susceptibility ($\propto U^2 \chi^{sp}_\omega$) separately shown with dotted red lines in all panels. The singlet-pairing term, which was a sizable contribution to the mild suppression of the diagonal elements of $\tilde{\chi}^{ch}_{\nu\nu'}$ in the high-$T$/perturbative regime is, instead, negligible in the local-moment regime. As expected, the low-frequency bubble contribution is almost vanishing in the HA. Differently, this term is sizable (though, generally smaller than in the perturbative regime) in the AIM or the DMFT solution of the Hubbard model, consistent with the Fermi-liquid nature of their ground states. In the latter two cases, we also notice the emergence of a non-negligible, albeit small, $U$-irr (multiboson) positive contribution at low frequencies. As a result, the low-frequency negative diagonal entries of the generalized susceptibility in the HA tend to exactly coincide with their spin contribution or more precisely with the part of it proportional to $\chi^{sp}_\omega$,

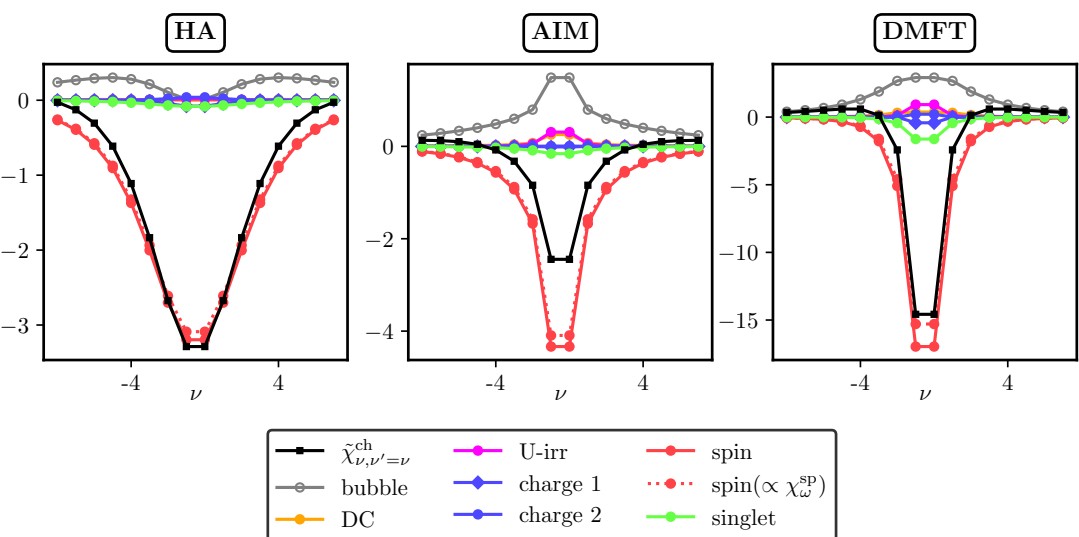

Figure 5: Full SBE decomposition of the diagonal entries of $\tilde{\chi}^{sp}_{\nu\nu'}$, as in Fig. 3, for the local moment regime ($T = 0.1, U = 5.75$ for HA and AIM and $T = 0.1, U = 2.2$ for the DFMT solution). For all three models, in this regime, we note an overwhelming predominance of the spin contribution (and in particular of its part proportional to $\chi^{sp}_\omega$) as damping effect of the diagonal entries of $\tilde{\chi}^{ch}_{\nu\nu'}$, which in the low-frequency domain considered prevail over the smaller positive contributions from the bubble term and of the multiboson-exchange processes.

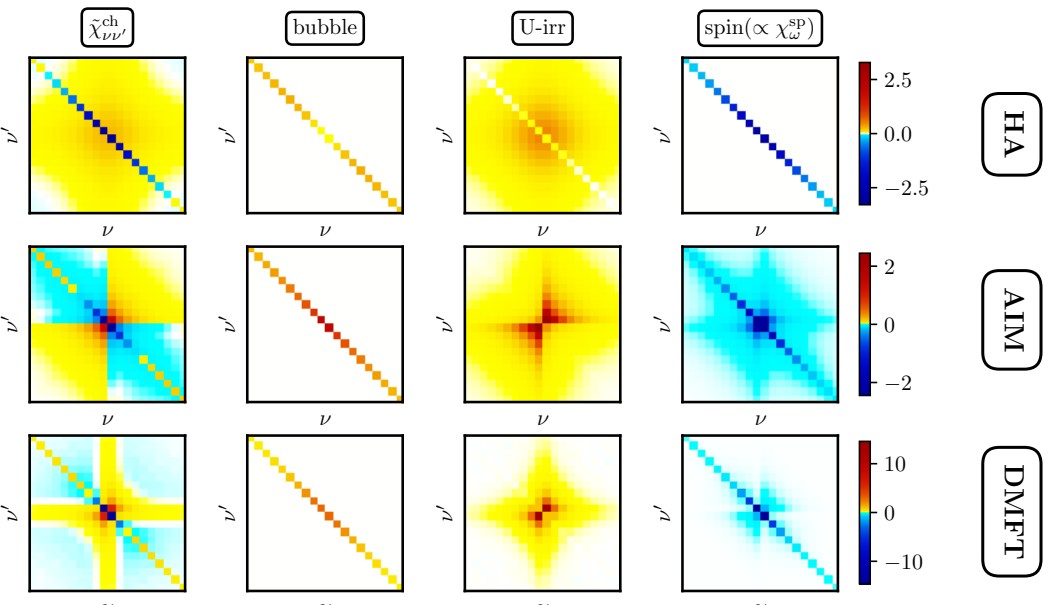

Figure 6: Results of our SBE decomposition of $\tilde{\chi}^{\mathrm{ch}}_{\nu\nu'}$ in the HA (top row), the AIM (center row) and the DMFT solution of the HM (bottom row) in the local moment regime (same parameters as Fig. 5). Similarly as in Fig. 4, the major contributions to $\tilde{\chi}^{\mathrm{ch}}_{\nu\nu'}$ in this regime are reported: the bubble term (second column panels), the multiboson processes (thrid column) as well as the predominant part of the spin, i.e. the one proportional to $\chi^{\mathrm{sp}}_{\omega}$. Differently of the perturbative case, in this parameter region the singlet contribution is negligible.

when $T$ is progressively lowered. In the other two models, the effect of the spin term, though largely predominant, is partially mitigated by the non-vanishing bubble contribution at low frequencies.

Guided by the scenario emerging from the SBE analysis of the diagonal entries of $\tilde{\chi}^{\mathrm{ch}}_{\nu\nu'}$, we broaden our study to the whole frequency domain. The corresponding numerical data are reported on the left-column panels of Fig. 6 similar to Fig 4. Beyond the negative low-frequency entries shown in Fig. 5, the generalized charge susceptibility displays off-diagonal elements, that significantly deviate from the faint negative background of the weak-coupling case, and are different for each model. Specifically, they build a faint positive background in the HA, totally reversing the overall sign structure of the low-frequency domain w.r.t. the perturbative regime, as mentioned in Sec. 2. In the AIM and DMFT case, instead, the off-diagonal frequency structures display different signs, seemingly interpolating between the weak-coupling and the HA case. In particular, the sign changes across the four quadrants for the AIM (where the same sign as in the perturbative calculations persists in the upper-left and lower-right quadrant, and gets flipped to the one of the HA in the other two), while a more complex cross-like structure emerges in the DMFT calculation of the HM.

Analogously to the previous subsection, in our SBE decomposition, we focus on the predominant contributions only. Consistent with the results discussed above for diagonal elements, these are the bubble term (second-column panels), the $U$-irr/multiboson (third column panels), and the spin contribution proportional to the physical susceptibility (rightmost-column panels). Indeed, the fact that these specific contributions correctly capture all main frequency structures of $\tilde{\chi}^{\mathrm{ch}}_{\nu\nu'}$ in the local moment regime of the three models can be readily appreciated in Fig. 6. More specifically, beyond the different behaviors of the (purely diagonal)

bubble terms in the three cases discussed above, two important observations can be made: (i) the suppressive effects arising from the spin contribution are by far the strongest along the diagonal; (ii) the multiboson processes tend to systematically increase the off-diagonal background.

In the perspective of our study, it is important to realize that (i) is a *direct* consequence of the local moment physics: When the local spin operator is a conserved quantity, as is the case in the HA, its (imaginary) time-dependence reduces to a (temperature-dependent) constant $[\chi^{\mathrm{sp}}(\tau) \equiv \langle S_z^2 \rangle]$, whose Fourier-transform yields a Kronecker-delta contribution for the (bosonic/transfer) Matsubara frequency, i.e., $\chi_\omega^{\mathrm{sp}} \equiv \beta \langle S_z^2 \rangle \delta_{\omega,0}$. This feature, which entails the Curie-law for the isolated moment of the HA, works as a strict frequency-selection rule $\nu = \nu'$ in the corresponding part of the spin SBE processes, cf. Eq. (11), thus explaining the purely diagonal shape of the spin contribution in the first row of Fig. 6. Similar considerations apply to the AIM and DMFT datasets, where in the local moment regime this is not strictly valid, but a good approximation. Here, the corresponding weak $\tau$ dependence of $\chi_\omega^{\mathrm{sp}}$ is reflected in a mild frequency broadening of the negative damping effects of the spin terms around the diagonal (cf. rightmost panels of the second and third row in Fig. 6). This clearly explains, in combination with (ii), why one observes sign changes in the off-diagonal background in the local moment regime of the AIM/DMFT, in contrast to the entirely positive background in the local moment regime of the HA. The sign changes in the AIM/DMFT data results from the interplay between the diffuse positive contribution of the multiboson processes, and the almost, but not perfectly diagonal damping effects of spin channel, while in the HA the damping effects do not occur anywhere else but for $\nu = \nu'$.

After identifying the spin term (and in particular, its part proportional to $\chi_\omega^{\mathrm{sp}}$) as the driving contribution to the significant, mostly diagonal, suppression effects in $\tilde{\chi}_{\nu\nu'}^{\mathrm{ch}}$, we inspect this contribution more closely. In particular, beyond the role of the large Curie/Curie-like static susceptibility $[\chi_{\nu'-\nu}^{\mathrm{sp}} \sim \beta \langle S_z^2 \rangle \delta_{\nu'-\nu,0}]$ discussed above, in Fig. 7 we report and analyze the individual roles played by the external legs $[G_\nu]^2[G_{\nu'}]^2$ and the Hedin spin-vertices $\lambda_{\nu,\nu'-\nu}^{\mathrm{sp}}\lambda_{\nu,\nu'-\nu}^{\mathrm{sp}}$ in suppressing the charge sector.

We start with the HA results, where, on the one hand, the Green's function product (leftmost panels) show an insulating behavior, i.e., its value is strongly reduced at low frequencies. On the other hand, the reduction is overcompensated by the square of the Hedin spin vertex (middle panel), which displays very large values at low-frequencies. This allows the strongly suppressive (rigorously diagonal in the HA) effect of the Curie susceptibility to be effectively transferred into the charge channel up to frequencies of order $U$.

A remarkable illustration of the interplay between suppressed $G_\nu$'s and enhanced Hedin vertices is provided by the derivation of a *quantitative* criterion for evaluating the size of the frequency/energy window associated to negative diagonal entries of $\tilde{\chi}_{\nu\nu'}^{\mathrm{ch}}$, and hence to non-perturbative effects, in the HA. Such a criterion, whose full derivation is reported in Appendix B, states that in the local-moment regime of the HA the value of the Hedin spin vertex should be

$$[\lambda_{\nu,0}^{\mathrm{sp}}]^2 \geq \frac{4}{3U^2}\frac{1}{|G_\nu|^2}, \qquad (12)$$

in order to get a sign change from positive to negative of a diagonal entry of $\tilde{\chi}_{\nu\nu'}^{\mathrm{ch}}$. Hence, deep in the local-moment regime of the HA, starting from a relatively large energy scale (e.g., $|\nu| \geq U$, where $\lambda_{\nu,0}^{\mathrm{sp}} \sim 1$ and $|G_\nu|^2 \sim \frac{1}{\nu^2}$) and gradually reducing $|\nu|$, one should expect the sign flip of $\tilde{\chi}_{\nu,\nu'=\nu}^{\mathrm{ch}}$ from positive to negative to occur at $|\nu| = |2n-1|\pi T \simeq \frac{\sqrt{3}}{2}U$. Indeed, this is *exactly* what happens [10, 25] in the low-$T$ limit of the HA. Such a quantitatively accurate SBE-based estimate, eventually accounting for the hitherto not understood prefactor of $\frac{\sqrt{3}}{2}$ recurrent in several preceding studies [1, 7, 10, 25] of the HA, pinpoints, once more, the *direct*

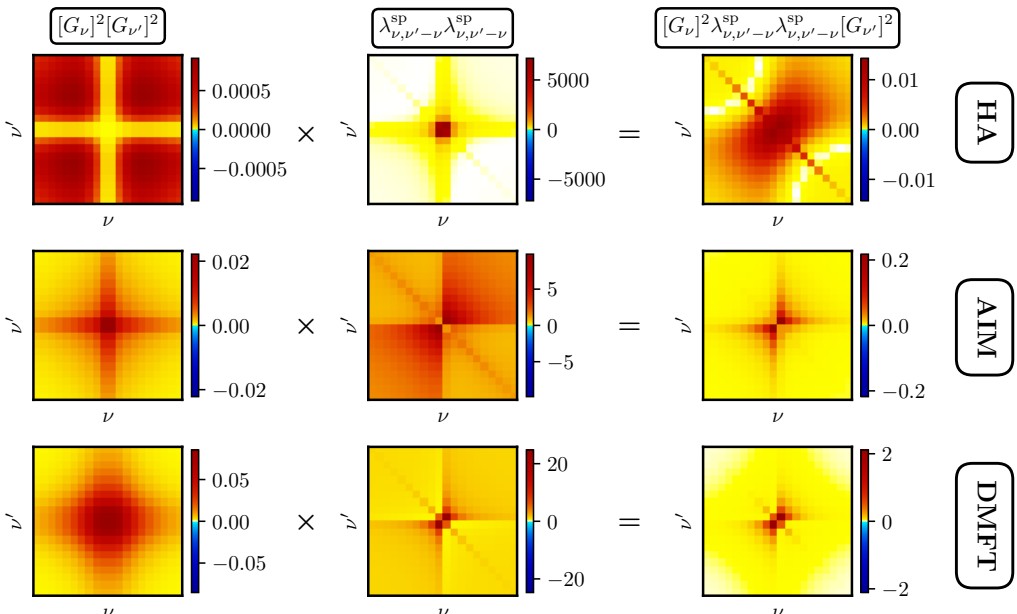

Figure 7: Partial decomposition of the spin contribution in terms of the Green's functions ($[G_\nu]^2[G'_\nu]^2$, left column) and the Hedin-spin vertex ($\lambda^{\mathrm{sp}}_{\nu,\nu'-\nu}$, center column) in the local-moment regime (same parameters as Fig. 5). The product of the first two columns is shown in rightmost column. $\tilde{\chi}^{\mathrm{sp}}_{\nu'-\nu}$ is not explicitly shown.

role of the enhanced spin exchange processes in suppressing local charge fluctuations and triggering the breakdown of the self-consistent perturbation theory in the local-moment regime. In particular, we note how the specific prefactor $\frac{\sqrt{3}}{2}$ naturally originates, in our SBE analysis, from the three possible spin orientations in the SU(2)-symmetric situation considered.

Turning back to the data of Fig. 7 and proceeding with the analysis of the AIM (second row) and of the DMFT results (third row), we note that, in contrast to the HA, the corresponding Green's function product tends to increase at the lowest frequencies. Consistent to our previous considerations, here the Hedin spin-vertex, while remaining systematically larger than its perturbative/asymptotic value ($\lambda^{\mathrm{sp}} \to 1$) is much smaller than in the HA case. A closer inspection of the results reveals that the largest values of the Hedin vertex are found in the lowest offdiagonal frequency domain. This feature is also reflected in the combined results for the product of Green's function and Hedin vertices, which is reported in the rightmost panels of Fig. 7. However, since the part of the spin contribution proportional to the physical susceptibility in the local-moment regime of the AIM/DMFT is not completely diagonal in contrast to the HA case, these relatively large off-diagonal effective couplings become now particularly important: They allow the on-site charge-fluctuation suppression to *also* affect the off-diagonal frequency domain of $\tilde{\chi}^{\mathrm{ch}}_{\nu\nu'}$. This can reduce or in some cases prevail over the overall positive contribution of the multiboson processes (cf. rightmost panels of Fig. 6).

Altogether, the analysis of the data in Fig. 7 underlines the importance of the interplay between the Green's function and the Hedin spin vertex. In fact, their products modulate frequency dependence and magnitude of the spin contribution, critically shaping the corresponding frequency structures of $\tilde{\chi}^{\mathrm{ch}}_{\nu\nu'}$ and their change from the perturbative to the non-perturbative regime. In particular, such products crucially control how the emergence of an enhanced response in the spin sector (here, e.g., the Curie one, associated to a local moment) can indeed affect the charge sector. This strong *intertwining* between different scattering sectors is evidently crucial in the non-perturbative regime, in order to correctly describe the physics of localized moments in *all* its aspects on an equal footing.

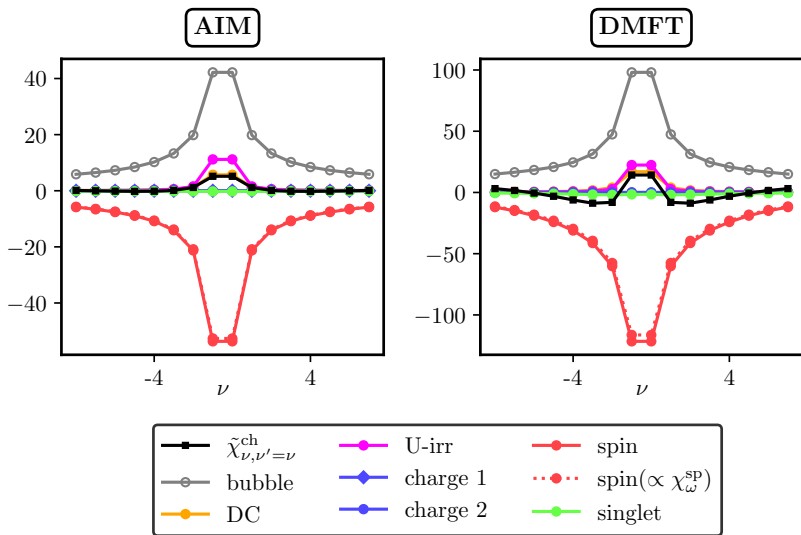

Figure 8: SBE decomposition of the diagonal elements of $\tilde{\chi}^{\mathrm{ch}}_{\nu\nu'}$ for the AIM and the DMFT solution of the HM, similar as in Figs. 3 and 5, but at lower temperatures, corresponding to the Kondo regime [25] of the AIM ($T = \frac{1}{60}, U = 5.75$) and of the DMFT solution ($T = 0.02, U = 2.2$). The major contributions have the same origin as in the local moment regime: a (positive) bubble term (gray dots), a (negative) spin contribution (red), specifically its part proportional to $\chi^{\mathrm{sp}}_{\nu'-\nu}$ (dashed red line), and a (positive, but smaller) U-irreducible term (magenta). In this parameter regime, a positive, albeit smaller, DC contribution (orange) can be also noted at the lowest frequencies.

### 3.3 Low-temperature/Kondo regime

Finally, we turn our attention to the low-temperature regime of the AIM and the DMFT solution of the HM , where the local moment is strongly screened, namely, at about the respective Kondo temperature (the HA, which does not feature such a regime, it is not considered in this section[8]).

The corresponding results for the diagonal elements of $\tilde{\chi}^{\mathrm{ch}}_{\nu\nu'}$ are presented in Fig. 8. At these (lower) temperatures (explicit values given in the caption), the diagonal of $\tilde{\chi}^{\mathrm{ch}}_{\nu\nu'}$ of both models is positive at the lowest Matsubara frequencies, turns slightly negative in a narrow intermediate frequency-window, eventually becoming positive again to match the expected high-frequency behavior. In Ref. [25], this particular frequency structure of the diagonal of $\tilde{\chi}^{\mathrm{ch}}_{\nu\nu'}$ (best seen here in the DMFT solution, right panel) was intuitively regarded as a characteristic hallmark of the Kondo-regime in the charge sector [25] and coined "onion-structure". According to our SBE decomposition, we note that similarly as for the local-moment regime (see Fig. 5) the two major positive and negative contribution are the bubble term and the spin term (and in particular, its part $\propto \chi^{\mathrm{sp}}_{\omega}$), respectively. However, as opposed to the local-moment regime, both contributions are now of similar size. Indeed, their balance, together with the smaller positive multiboson (and, to a lesser extent, DC) contribution at the lowest frequencies, features the characteristic sign changes along the diagonal and, hence, the onion-structure.

---

[8]Evidently, for the HA, the considerations made for its local-moment regime will simply apply even more strictly by further reducing the temperature.

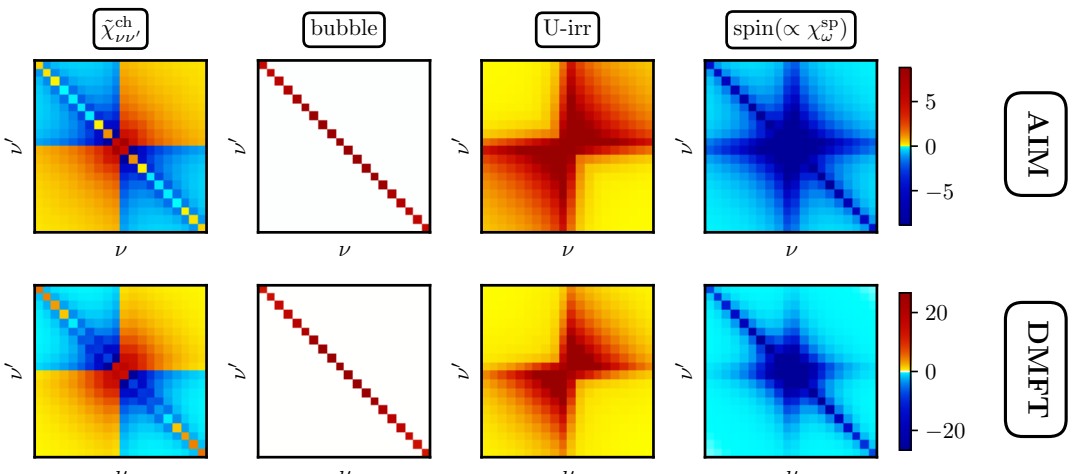

Figure 9: Results of our SBE decomposition of $\tilde{\chi}^{\text{ch}}_{\nu\nu'}$ the AIM (top row) and the DMFT solution of the HM (bottom row) in the Kondo regime (same parameters as Fig. 8). Both models feature a very similar structure of $\tilde{\chi}^{\text{ch}}_{\nu\nu'}$ (left panels), clearly displaying the characteristic onion-structure associated [25] to the Kondo regime. Similarly as in Fig. 6, the major contributions to $\tilde{\chi}^{\text{ch}}_{\nu\nu'}$ in this regime are reported: the bubble term (second column panels), the multiboson processes (third column) as well as the predominant part of the spin SBE, i.e. the one proportional to $\chi^{\text{sp}}_{\omega}$. In this parameter region, however, the (positive) bubble and multiboson contributions are much larger than in the local moment regime, while the negative spin term, albeit still large, is no longer essentially localized on the diagonal $\nu = \nu'$. The emergence of the onion-structure appears, thus, a direct consequence of the sensitive balance between these three terms of comparable magnitude.

As for the full matrix structure of $\tilde{\chi}^{\text{ch}}_{\nu\nu'}$, shown in Fig. 9 (leftmost panels), one easily recognizes both the aforementioned onion-structure on the diagonal, as well as an evident quadrant-dependent structure of all off-diagonal terms. Consistent with the analysis of Fig. 8, our SBE decomposition indicates that the whole frequency structure of $\tilde{\chi}^{\text{ch}}_{\nu\nu'}$ in the Kondo regime remains essentially controlled by the same terms as in the local-moment regime, namely the bubble, the spin (the part proportional to $\chi^{\text{sp}}_{\omega}$) and the multiboson term, whose contributions are explicitly shown in the second-to-fourth column panels. These terms, however, have different magnitudes (and in the case of the spin term also a different structure) than in the local moment regime, yielding comparable values. While we have already associated the onion-structure to the balance of these three terms, the quadrant structure clearly originates from the competition of the positive multiboson and of the negative spin term. Specifically, the former term displays stronger intensities in the second and fourth quadrant, while the latter one, due to the screening and life-time reduction of the local moment, is now acting on a relatively large symmetric frequency interval $\nu - \nu'$ around the diagonal.

Since the change of the spin contribution w.r.t. to the local moment regime emerges as a decisive ingredient in shaping the overall frequency behavior of $\chi^{\text{ch}}_{\nu\nu'}$ and, in particular, its onion-structure, it is worth to investigate it more thoroughly. To this end, in Fig. 10 we report an analysis of the coupling of the spin channel, similarly as that in Fig. 7. The main difference w.r.t. the local moment regime is the *suppression* instead of the increase of the Hedin spin-vertices $[\lambda^{\text{sp}}_{\nu,\nu'-\nu}]^2$ for the lowest diagonal frequency elements. In fact, these elements (represented in Fig. 10 by the yellow colors) reach values *below* the non-interacting/high-frequency limit of 1. Rather, the offdiagonal sectors of $[\lambda^{\text{sp}}_{\nu,\nu'-\nu}]^2$ as well as the Green's functions show

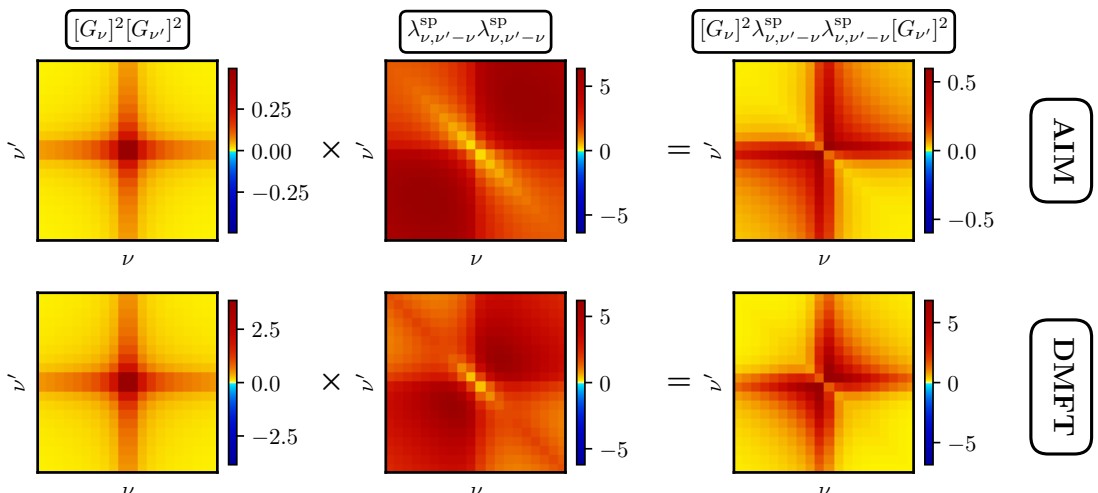

Figure 10: Similarly as in Fig. 7 but for low-temperatures, in the Kondo regimes of the respective models (see Fig. 8 for the exact values).

a qualitatively similar behavior as in the local moment case, in particular for the AIM. Altogether, this leads to a reduction of the spin contribution, which shifts the balance between the spin, the multiboson and the "bubble" term in favor of the latter two.

The main core of our findings, i.e. the predominant role played by the static spin response and the electron-spin scattering amplitude in shaping the generalized charge susceptibility in the local moment/Kondo screening regimes, can be further supported by considering *ad hoc* approximations, which neglect and/or simplify specific terms in the full SBE decomposition of Eq. 11. This systematic study is presented in Appendix C and D. The results discussed there convincingly demonstrate that the single spin exchange and the multiboson processes are driving the qualitative changes of the generalized charge susceptibility w.r.t. the bubble term. Further, the comparison between different approximation schemes, in particular with those where the spin-electron coupling $\lambda^{\mathrm{sp}}$ is kept fixed to its perturbative/asymptotic value of 1, highlights once more the crucial importance of its frequency dependence for capturing the intrinsic link between local charge and spin fluctuations in the non-perturbative regimes of the many-electron problem.

## 4 Conclusion

In this work, we have studied the microscopic mechanisms controlling the different Matsubara frequency structures displayed by the generalized charge response $\tilde{\chi}^{\mathrm{ch}}_{\nu\nu'}$ [25] in relevant regimes of many-electron physics, from the high-temperature perturbative regime, to the local moment and the Kondo regime. To this aim, we have computed $\tilde{\chi}^{\mathrm{ch}}_{\nu\nu'}$ of three significant models, namely the HA, the AIM and the HM on the Bethe lattice solved within DMFT, for decreasing temperature values, corresponding to the physical situations mentioned above, and decomposed it in terms of its single-boson exchange (SBE) contributions. By exploiting the applicability of the SBE decomposition in all parameter regimes, including the non-perturbative region, we were able to precisely identify the microscopic mechanisms shaping the frequency dependence of $\tilde{\chi}^{\mathrm{ch}}_{\nu\nu'}$.

In particular, as discussed in Sec. 2, we were interested to investigate *why* the suppression of the on-site charge fluctuations, which constitutes an essential aspects of the local moment formation, occurs in the specific way it was observed [3,16,17,25,28,35,43] in several funda-

mental many-electron models. Our study has demonstrated that, in the local moment regime, the strongly negative diagonal entries of $\tilde{\chi}^{\text{ch}}_{\nu\nu'}$, which appear at low frequencies and drive the suppression of local charge fluctuations, directly originate from the exchange of static ($\omega = 0$) spin fluctuations.

Remarkably, while the magnitude of such spin-driven suppression effects gets enhanced by *both* the Curie-behavior of the magnetic response *and* the increased spin-fermion coupling, their action is overwhelmingly concentrated for $\nu = \nu'$ as a *precise consequence* of the long lifetime of the local moment in this parameter regime. Instead, the off-diagonal background emerges from the competition between the residual (negative/suppressive) contribution of the spin exchange for $\nu \neq \nu'$, and the overall and smooth frequency-dependent positive contribution of the multiboson processes (in the HA, where the local spin corresponds to a rigorously conserved quantity, the low-frequency background is, in fact, entirely positive). In this way, our analysis eventually provides the "smoking gun" evidence of the direct link existing between the frequency structures of $\tilde{\chi}^{\text{ch}}_{\nu\nu'}$ which trigger the *breakdown* of the self-consistent perturbation expansion and the physical properties of the long-lived magnetic moment, validating the heuristic interpretation [25] of the negative diagonal entries of $\tilde{\chi}^{\text{ch}}_{\nu\nu'}$ as its "fingerprint".

More precisely, our study clarifies why any consistent description of the physics associated to the existence of a local moment in all scattering sectors (i.e., where the enhanced on-site magnetic response matches a simultaneous freezing of on-site charge and pairing fluctuations) is intrinsically *nonperturbative* due to the intertwining properties of the scattering processes in the different channels.

Our analysis of the Kondo/low-temperature regime of the AIM and the HM solved in DMFT appears also fully consistent with the above interpretation. The partial revival of charge fluctuations observed by lowering the temperature can be mostly ascribed to a significant reduction of the Hedin spin-fermion vertex $\lambda^{\text{sp}}_{\nu,\nu-\nu'}$, which becomes even smaller than its perturbative value of 1 for low frequencies. Such a low-frequency "decoupling" of the spin SBE processes from the charge sector significantly mitigates the corresponding suppression effect on the lowest frequency diagonal entries featuring the corresponding onion-structure [25] of $\tilde{\chi}^{\text{ch}}_{\nu\nu'}$. At the same time, the shorter life-time of the Kondo screened moment is reflected in much more diffuse negative damping effects for $\nu \neq \nu'$, featuring the characteristic sign-quadrant structure of the off-diagonal background, via a more balanced competition with the positive contribution of the multiboson processes.

These conclusions are supported by the analysis of the approximations obtained by neglecting specific contributions of the SBE decomposition as well as by setting $\lambda^{\text{sp}}_{\nu,\nu-\nu'} = 1$. In fact, we managed to demonstrate that the combination of the bubble, the U-irreducible and the spin term indeed provides a minimal set to qualitatively capture the main frequency structures of $\tilde{\chi}^{\text{ch}}_{\nu\nu'}$ for all temperatures considered, with an improved match in the non-perturbative, local moment regime.

Our precise identification of the predominant microscopical processes shaping the generalized charge response of many-electron models in different physical regimes, and of the fundamental role played by the strong intertwining of spin and charge fluctuations in triggering the breakdown of the self-consistent perturbation expansion, suggests future applications of the SBE decomposition procedure to other non-perturbative situations of fundamental interest. For instance, it would be interesting to extend our analysis to the case of attractive (negative-$U$) interactions, where the role of the "bound-state" responsible for non-perturbative effects [20, 66, 67] will be played by the (pre-)formation of local Cooper pairs, instead of local magnetic moments. On a broader perspective, an extension of our SBE study to the nonlocal fluctuations of two-dimensional many-electron systems, e.g. of those captured by diagrammatic Monte Carlo [68, 69] or cluster/diagrammatic extensions of DMFT [39, 70] for the HM, may provide interesting new information on the non-perturbative intertwining of commensu-

rate/incommensurate antiferromagnetic [71] and charge/pairing fluctuations, shedding new light on the highly debated topic of unconventional superconductivity and stripe formation in different parameter regimes [72–77].

## Acknowledgments

We would like to thank S. Andergassen, P. Bonetti, M. Capone, S. Ciuchi, K. Fraboulet, E. Kozik, E. van Loon, L. Del Re, M. Reitner, G. Rohringer, T. Schäfer, and D. Vilardi for the valuable exchanges of ideas. Calculations were performed on the Vienna Scientific Cluster (VSC).

**Funding information**   This work was supported by the Austrian Science Fund (FWF) through project I 5487 (A. T. and P. C.-G.), project I 5868 (Project P01, part of the FOR 5249 [QUAST] of the German Research Foundation, DFG) (S.A.) and through the DFG FOR 5249 [QUAST] - 449872909 (Project P05) (G.S.). S.A. and G.S. also acknowledge support from the Deutsche Forschungsgemeinschaft (DFG) through the Würzburg-Dresden Cluster of Excellence on Complexity and Topology in Quantum Matter-ct.qmat (EXC 2147, project-id 390858490).

## A   Definitions

In order to provide a concise main material, some specific definitions needed in Sec. 2.3 were moved to this appendix.

Formally the physical susceptibilities in Eq. (9) can be written as [29,38]:

$$\chi_\omega^{\text{ch/sp}} = -\frac{2}{\beta}\sum_\nu G_\nu G_{\nu+\omega} - \frac{2}{\beta^2}\sum_{\nu\nu'} G_\nu G_{\nu+\omega} F_{\nu\nu',\omega}^{\text{ch/sp}} G_{\nu'} G_{\nu'+\omega}\,,\tag{A.1}$$

and

$$\chi_\omega^{\text{si}} = \frac{1}{\beta}\sum_\nu G_\nu G_{\omega-\nu} - \frac{1}{2\beta^2}\sum_{\nu\nu'} G_\nu G_{\omega-\nu} F_{\nu\nu',\omega}^{\text{si}} G_{\nu'} G_{\omega-\nu'}\,,\tag{A.2}$$

where the singlet vertex function $F_{\nu\nu',\omega}^{\text{si}}$ can be calculated from:

$$F_{\nu\nu',\omega}^{\text{si}} = \frac{1}{2}(F_{\nu\nu',\omega-\nu-\nu'}^{\text{ch}} - 3F_{\nu\nu',\omega-\nu-\nu'}^{\text{sp}})\,.\tag{A.3}$$

Further the Hedin vertices can be calculated from the full vertex $F$ in the following way [61]:

$$\lambda_{\nu,\omega}^{\text{ch/sp}} = \frac{1 + \sum_{\nu'} F_{\nu\nu'\omega}^{\text{ch/sp}} G_{\nu'} G_{\nu'+\omega}}{w_\omega^{\text{ch/sp}}/U^{\text{ch/sp}}}\,,\tag{A.4}$$

$$\lambda_{\nu,\omega}^{\text{si}} = \frac{-1 + \frac{1}{2}\sum_{\nu'} F_{\nu\nu'\omega}^{\text{si}} G_{\nu'} G_{\omega-\nu'}}{w_\omega^{\text{si}}/U^{\text{si}}}\,.\tag{A.5}$$

## B   Derivation of negativity criterion

In the following we derive eq. 12. The idea behind this approximation of $\chi_{\nu\nu'}^{\text{ch}}$ is to determine at which Matsubara frequency negative elements on the diagonal of $\chi_{\nu\nu'}^{\text{ch}}$ appear in the Hubbard

atom and to give a simplified criterion. Starting from the definition of the full vertex (see Eq. 9) [29],

$$F^{\mathrm{ch}}_{\nu\nu'\omega} = \phi^{\mathrm{Uirr,ch}}_{\nu\nu'\omega} + \lambda^{\mathrm{ch}}_{\nu\omega} w^{\mathrm{ch}}_{\omega} \lambda^{\mathrm{ch}}_{\nu'\omega} - \frac{1}{2} \left( \lambda^{\mathrm{ch}}_{\nu,\nu-\nu'} w^{\mathrm{ch}}_{\omega} \lambda^{\mathrm{ch}}_{\nu+\omega,\nu-\nu'} + 3\lambda^{\mathrm{sp}}_{\nu,\nu'-\nu} w^{\mathrm{sp}}_{\nu'-\nu} \lambda^{\mathrm{sp}}_{\nu+\omega,\nu'-\nu} \right)$$
$$+ \frac{1}{2} \lambda^{\mathrm{si}}_{\nu,\nu+\nu'} w^{\mathrm{si}}_{\nu+\nu'+\omega} \lambda^{\mathrm{si}}_{\nu',\nu+\nu'+\omega} - 2U\,, \tag{B.1}$$

we consider the static limit i.e. $\omega = 0$. For $U \gg T > T_{\mathrm{Kondo}}$ the charge susceptibility $\chi^{\mathrm{ch}}_{\omega}$ is negligible and, therewith, also the singlet susceptibility $2\chi^{\mathrm{si}}_{\omega} = \chi^{\mathrm{ch}}_{\omega}$ due to particle-hole symmetry. Then, using the definition of $w^{\alpha}_{\omega}$ from Eq. 10, we write $\lambda^{\mathrm{ch}} w^{\mathrm{ch}} \lambda^{\mathrm{ch}} \approx U$ and $\lambda^{\mathrm{si}} w^{\mathrm{si}} \lambda^{\mathrm{si}} \approx 2U$, which cancels the double counting correction in Eq. B.1, yielding

$$F^{\mathrm{ch}}_{\nu\nu',\omega=0} = \phi^{\mathrm{Uirr,\,ch}}_{\nu\nu',\omega=0} - \frac{3}{2} \lambda^{\mathrm{sp}}_{\nu,\nu'-\nu} w^{\mathrm{sp}}_{\nu'-\nu} \lambda^{\mathrm{sp}}_{\nu,\nu'-\nu} - \frac{1}{2} U\,, \tag{B.2}$$

for the vertex function.

From here on, we turn our attention to the generalized charge susceptibility and focus on the diagonal of $\chi^{\mathrm{ch}}_{\nu\nu'}$ only, formulating an inequality defining a criterion for its values to be smaller than zero,

$$\tilde{\chi}^{\mathrm{ch}}_{\nu,\nu'=\nu} \le 0\,, \tag{B.3}$$

$$-\beta[G_\nu]^2 - [G_\nu]^2 (\phi^{\mathrm{Uirr}}_{\nu=\nu'} - \frac{3}{2} \lambda^{\mathrm{sp}}_{\nu,\nu'-\nu=0} w^{\mathrm{sp}}_{\nu'-\nu=0} \lambda^{\mathrm{sp}}_{\nu,\nu'-\nu=0} - \frac{1}{2} U)[G_\nu]^2 \le 0\,. \tag{B.4}$$

Assuming further that the dominant contribution stems from the spin channel and thus neglecting the remaining DC and U-irreducible contribution this simplifies to

$$-\beta[G_\nu]^2 - \frac{3}{4}[G_\nu]^2 \lambda^{\mathrm{sp}}_{\nu,0} U^2 \chi^{\mathrm{sp}}_{\omega=0} \lambda^{\mathrm{sp}}_{\nu,0}[G_\nu]^2 \le 0\,. \tag{B.5}$$

At the low temperatures considered here $\chi^{\mathrm{sp}}_{\omega=0}$ follows the Curie-Weiss law and, therefore, $\chi^{\mathrm{sp}}_{\omega=0} \sim \beta$

$$\underbrace{-\beta[G_\nu]^2}_{>0} \left( 1 + \frac{3}{4} \lambda^{\mathrm{sp}}_{\nu,0} U^2 \lambda^{\mathrm{sp}}_{\nu,0}[G_\nu]^2 \right) \le 0\,. \tag{B.6}$$

$$1 + \frac{3}{4} \lambda^{\mathrm{sp}}_{\nu,0} U^2 \lambda^{\mathrm{sp}}_{\nu,0}[G_\nu]^2 \le 0\,, \tag{B.7}$$

which yields, taking into account that $G_\nu$ is purely imaginary, the criterion given in Eq. (12):

$$[\lambda^{\mathrm{sp}}_{\nu,0}]^2 \ge \frac{4}{3U^2 |G_\nu|^2}\,. \tag{B.8}$$

## C  Insights from approximations

In Sec. 3 we were able to identify the crucial role played by the spin contribution (see Eq. (11)) for the significant changes in the Matsubara frequency structure of the generalized charge susceptibility throughout the different temperature regimes. The validity of our interpretation is further supported by the study presented in this appendix, where we approximate the SBE decomposition to various extents and analyze the resulting $\tilde{\chi}^{\mathrm{ch}}_{\nu\nu'}$. The results of this analysis are shown in Fig. 11, where the upper three rows display data for the AIM in all three temperature regimes, and the bottom row displays it for the HA only in the local-moment regime. In the

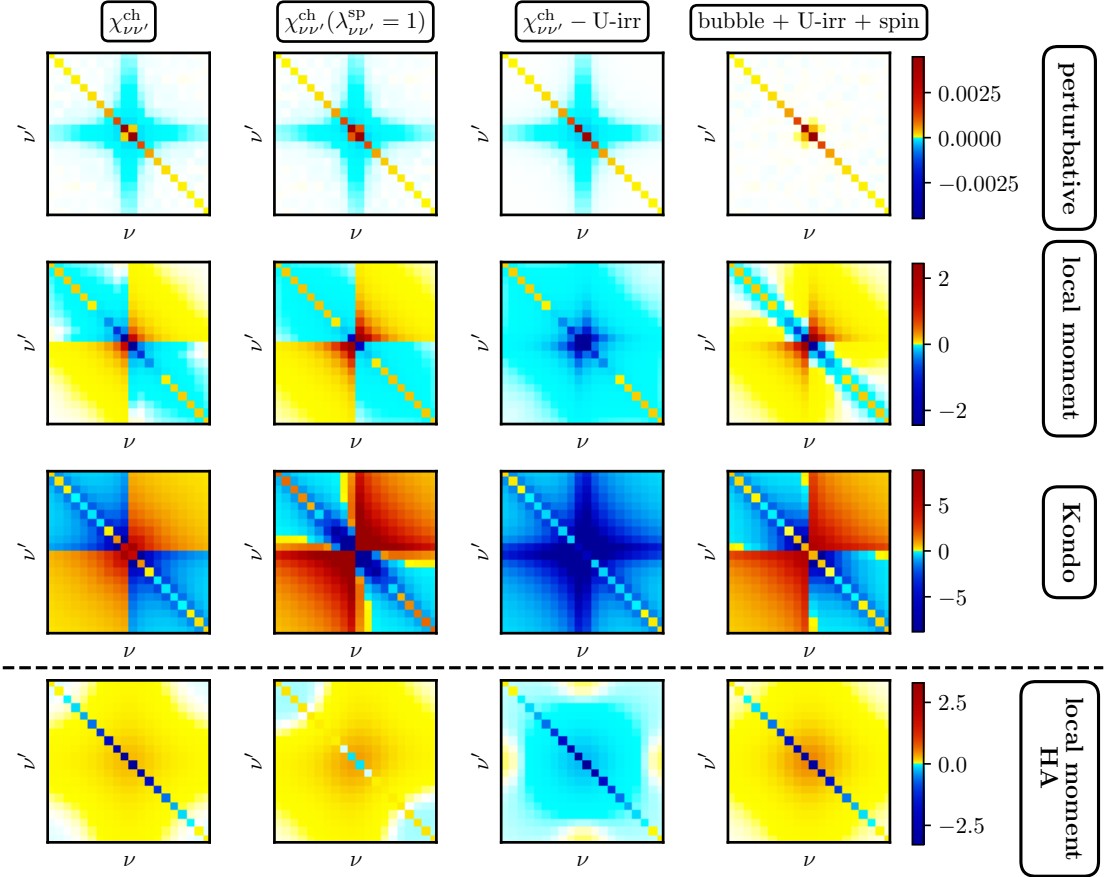

Figure 11: Different approximations of $\tilde{\chi}^{\text{ch}}_{\nu\nu'}$ for the AIM in all three temperature regimes (first three rows) and the HA at the local moment regime (fourth row). The first column shows the exact $\tilde{\chi}^{\text{ch}}_{\nu\nu'}$, the second column $\tilde{\chi}^{\text{ch}}_{\nu\nu'}$ when using the approximation: $\lambda^{\text{sp}}_{\nu,\nu'-\nu} = 1 \, \forall \, \nu, \nu'$, i.e. its non-interacting limit, the third column shows $\tilde{\chi}^{\text{ch}}_{\nu\nu'}$ without the U-irreducible channel and the fourth column displays the sum of the most important contributions (bubble, U-irreducible and spin).

leftmost panels of Fig. 11 the original, fully frequency-dependent, $\tilde{\chi}^{\text{ch}}_{\nu\nu'}$ is reported, as shown above in Figs. 4,6 and 9.

As a first step, the role of the Hedin spin vertex is inspected, by analyzing the resulting $\tilde{\chi}^{\text{ch}}_{\nu\nu'}$, if $\lambda^{\text{sp}}_{\nu,\nu'-\nu}$ is set to its perturbative/high frequency value of one ($\lambda^{\text{sp}}_{\nu,\nu'-\nu} = 1 \, \forall \, \nu, \nu'$). In particular, the results shown in the second column of Fig. 11 ,demonstrate that the frequency dependence of $\lambda^{\text{sp}}_{\nu,\nu'-\nu}$ is important to correctly capture the main non-perturbative structures in the generalized charge susceptibility: the "fingerprint" of the local moment is to a large extend washed away and the onion-structure is no longer visible when setting $\lambda^{\text{sp}}_{\nu,\nu'-\nu} = 1$. Overall, this analysis demonstrates the increasing importance of $\lambda^{\text{sp}}_{\nu,\nu'-\nu}$ as the temperature is lowered. For both, the HA and the AIM (similarly for the HM, not shown) the magnitude of the Hedin vertex $\lambda^{\text{sp}}_{\nu,\nu'-\nu}$ proves to be crucial for the delicate interplay among the different SBE decomposition terms, which eventually determines the sign structure of $\tilde{\chi}^{\text{ch}}_{\nu\nu'}$ and drives the breakdown of the perturbative expansion.

In order to further support the insights gained into the origin of the quadrant structure of $\tilde{\chi}^{\text{ch}}_{\nu\nu'}$, in the third column of Fig. 11 the U-irreducible term of Eq. (11) is neglected. As the results at intermediate and low-temperatures clearly demonstrate, the U-irreducible channel is crucial to properly describe the correct quadrant structure of $\tilde{\chi}^{\text{ch}}_{\nu\nu'}$. Specifically, the positive

quadrants are *absent* without the positive multiboson contribution. To a small amount this is already visible in the high-temperature regime, where the first two off-diagonal Matsubara frequencies have the wrong sign in this approximation. Evidently, since the HA has no quadrant structure as the AIM, neglecting the U-irreducible contribution affects all quadrants here. Additionally, neglecting the positive contribution of the U-irreducible channel also affects the onion-structure found at low temperatures in the AIM. In fact, the positive diagonal elements at lowest Matsubara frequencies are absent, which further underlines the importance of the multiboson term in this region, where the bubble and spin contribution almost cancel each other (cf. Fig. 8).

Further, we challenge the insights gained so far, by taking only those terms of the SBE decomposition into account, which were identified in the previous section to be most relevant for the sign structure and overall frequency dependence of the generalized susceptibility: the bubble term, the fully U-irreducible contribution and the spin channel. The results of this approximation are shown in the last column of Fig. 11. Overall, a more than satisfactory reproduction of the most relevant qualitative frequency structures of $\tilde{\chi}_{\nu\nu'}^{\text{ch}}$ is found. Naturally, quantitative and small qualitative differences to the exact generalized susceptibility are present throughout all temperature regimes. In particular, in the perturbative regime, the negative background is missing (due to the missing singlet term, see Sec. 3.1) and in the Kondo regime, the central diagonal elements are positive, but their magnitude is too small. On the other hand, the local moment regime of both, the AIM and the HA is captured particularly well, in agreement with the discussion and the analysis of Sec. 3.2.

As a last step, we highlight the importance of the Hedin vertices by inspecting more closely the diagonal entries of the SBE spin contribution in Fig. 12 (the corresponding study for the full frequency structure is presented in Appendix D). The three leftmost panels show the results obtained in the perturbative, local moment and Kondo regime of the AIM, whereas the rightmost panel those for the local moment regime of the HA. The red dotted lines with circles correspond to the full contribution proportional to $\chi^{\text{sp}}(\omega)$ (similar to Fig. 3, 5 and 8), while orange dotted lines with triangles correspond to the same contribution under the approximation that $\lambda_{\nu,\nu'-\nu}^{\text{sp}} = 1$. In the perturbative regime of the AIM there is almost no difference between the full and the approximated contribution. Lowering the temperature to the local moment regime, we see an increased importance of $\lambda_{\nu,\nu'-\nu}^{\text{sp}}$. Neglecting its change from the non-perturbative limit leads to an underestimation of the spin contribution. This effect is par-

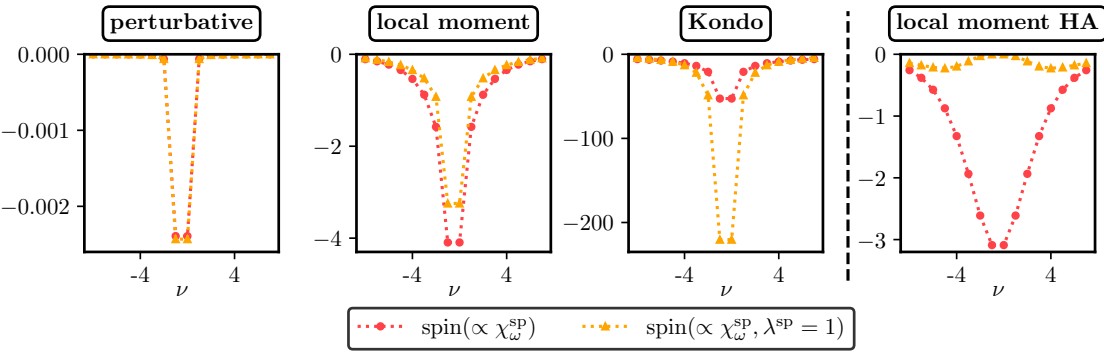

Figure 12: Frequency diagonal contributions to the generalized charge susceptibility of Eq. (11) arising from the spin channel (specifically: the part $\propto \chi^{\text{sp}}(\omega)$) in the perturbative, local moment and Kondo regime of the AIM as well as the local moment regime of the HA (red dotted line with circles) compared to the corresponding contribution obtained under approximation that $\lambda_{\nu,\nu'-\nu}^{\text{sp}} = 1$ (orange dashed line with triangles).



ticularly evident in the local moment regime of the HA, where the Hedin spin vertex has much larger values cf. Fig. 7.

Finally, we focus on to the Kondo regime of the AIM, where the frequency structure of $\lambda^{\text{sp}}_{\nu,\nu'-\nu}$ is such that, in contrast to the local moment regime, approximating it to unity results in a significant overestimation of the spin contribution. In fact, it is precisely the suppression of the full spin contribution due to the non-trivial frequency structure of the Hedin spin vertices that mostly drives the emergence of the characteristic onion structure in $\chi^{\text{ch}}_{\nu,\nu'-\nu}$. The latter appears indeed to be lost, when $\lambda^{\text{sp}}_{\nu,\nu'-\nu}$ is approximated to 1 cf. Fig. 11.

## D  Approximating $\lambda^{\text{sp}}_{\nu,\nu'-\nu} = 1$

Complementing our discussion in Appendix C we present the full frequency structure of the SBE spin contribution proportional to $\chi^{\text{sp}}(\omega)$ $\left(\text{i.e. } \frac{3}{2}[G_\nu]^2 \lambda^{\text{sp}}_{\nu,\nu'-\nu} U^2 \chi^{\text{sp}}_{\nu'-\nu} \lambda^{\text{sp}}_{\nu,\nu'-\nu} [G_{\nu'}]^2 \right)$ to the generalized charge susceptibility obtaining with and without approximating $\lambda^{\text{sp}}_{\nu,\nu'-\nu}$ in Fig. 13. In the perturbative regime of the AIM (first column) the corresponding frequency structure is barely affected. However, lowering the temperature to the local moment regime (second column) not only the diagonal is slightly suppressed, but also the off-diagonal components. This, of course, does not apply to the case of the HA (fourth column), where, since the on-site spin is a conserved quantity, no off-diagonal frequency structure is present in the corresponding contribution. The situation appears partially reversed in the Kondo regime (third column), where, the diagonal entries of the spin-contribution get enhanced when approximating $\lambda^{\text{sp}}_{\nu,\nu'-\nu}$ with its non-interacting limit and the off-diagonal is suppressed.

Overall, approximating $\lambda^{\text{sp}}_{\nu,\nu'-\nu} = 1$ mainly affects the diagonal of the spin contribution. However, this represents one of the most important ingredients driving the suppression of the

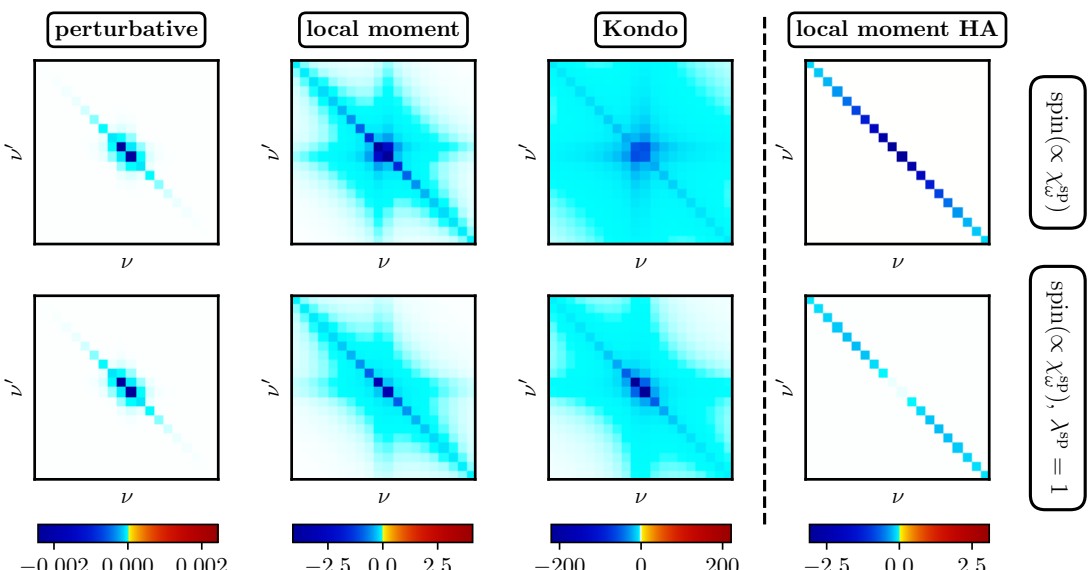

Figure 13: Frequency structure of the spin contribution of Eq. 11 (only the part proportional to $\chi^{\text{sp}}(\omega)$) for the perturbative (first column), local moment (second column) and Kondo regime of the AIM (third column) and the local moment regime of the HA (fourth column). The first row shows the correct frequency structure, the second row shows the frequency structure obtained under the approximation $\lambda^{\text{sp}}_{\nu,\nu'\nu} = 1$.

physical charge response in the local moment regime of the AIM and the revival of the same in the Kondo regime.

## E    Evolution of the physical spin and charge response function with temperature

In this Appendix we discuss the specific temperature values chosen for the analysis of the AIM and of the HA in relation to the overall $T-$behavior of the physical spin and charge response. In Fig. 14, $T\chi^{\mathrm{sp}}(\omega=0)$ and $\chi^{\mathrm{ch}}(\omega=0)$ as a function of T are shown, while the corresponding temperatures used to characterize the Kondo (K), the local-moment (LM) and the perturbative (P) physics throughout this work are marked with arrows. Let us note that the general behavior of $T\chi^{\mathrm{sp}}(\omega=0)$ and $\chi^{\mathrm{ch}}(\omega=0)$ for the AIM is well-known and well-documented, also in standard textbooks, such as e.g. [42,78]. For the sake of completeness, we briefly summarize the observations most relevant for our work here. The upper panels show the results for the AIM (green solid line with diamonds), where the effects of the local moment formation and the Kondo screening can be seen in relatively weak $T$-dependence of $T\chi^{\mathrm{sp}}(\omega=0)$ around its maximum (reminiscent of a Curie-behavior), its subsequent suppression as well as the clear minimum of $\chi^{\mathrm{ch}}(\omega=0)$ at low temperatures [43]. This behavior is best understood by com-

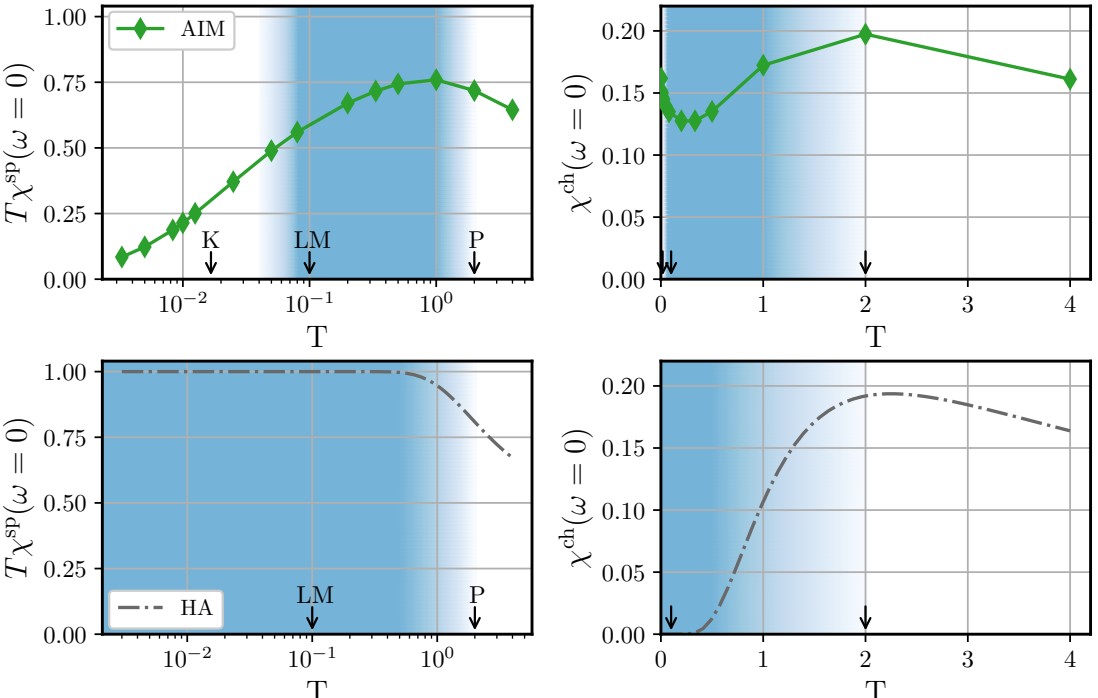

Figure 14: Temperature dependence of $T\chi^{\mathrm{sp}}(\omega=0)$ for the AIM (left upper panel) and the HA (left lower panel) plotted on a logarithmic scale and temperature dependence of $\chi^{\mathrm{ch}}(\omega=0)$ for the AIM (right upper panel) and the HA (right lower panel) on a linear scale. The blue-shadowed areas are shown as a mere guide to the eye to locate the parameter regimes where local moment physics is expected in the different models. The black arrows mark the location of the respective temperature regimes of the main text (K=Kondo screened, LM=Local Moment, P=Perturbative), s. text right before Sec. 3.1. Partially redrawn from [54].

paring it with the results for the HA (gray dotted dashed line in the lower panels), where an almost perfect plateau in $T\chi^{\mathrm{sp}}(\omega=0)$ and a concomitant monotonic suppression of $\chi^{\mathrm{ch}}(\omega=0)$ are observed, due to the absence of the Kondo screening in this model.

It is worth also noticing that all the temperatures chosen for the LM corresponds to parameter sets well inside the borders of the corresponding LM regions, and that this also applies to the DMFT case, e.g., when considering the different definitions of the crossover borders to the LM regime, as defined in the recent literature [28, 53]. In this respect, we observe that, due to the large prevalence of a low-frequency suppression of the diagonal entries of $\tilde{\chi}^{\mathrm{ch}}_{\nu\nu'}$ in the LM regime, the shape of the region where such suppression overcomes a given threshold (for instance, where its lowest diagonal element becomes negative) will match, though only at a rough qualitative level, the different criteria used and compared in [28, 53]. In particular, by taking a closer look at the data of Fig. 3 in the Supplemental Material of [25], as well as to the results presented in more details in [54], one can readily evince that the appearance of the first negative diagonal element of [25] would define a similar border to the LM regime as the "fingerprint criterion" shown in Ref. [53], only slightly shifted towards larger values of $U$. Specifically, at high $T$ the crossover border will be confined between the first and the second vertex divergence of the two-particle irreducible vertex in the charge sector, while at low-temperature between the "fingerprint" criterion [25] for $T_K$ ($\tilde{\chi}^{\mathrm{ch}}_{\pi T,\pi T} = \tilde{\chi}^{\mathrm{ch}}_{\pi T,-\pi T}$) and the qualitatively similar condition of $\tilde{\chi}^{\mathrm{ch}}_{\pi T,\pi T} = -\tilde{\chi}^{\mathrm{ch}}_{\pi T,-\pi T}$.

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
