# Peer review of "Non-perturbative intertwining between spin and charge correlations: A "smoking gun" single-boson-exchange result"

_SciPost Physics, doi:SciPost Phys. 16, 054 (2024)_

## Round 2 · Referee Report · Anonymous (Referee 1) · 2023-3-11

Strengths

The paper "Non-perturbative intertwining between spin and charge correlations: A "smoking gun" single-boson-exchange result" studies the interrelation between local spin and charge correlations in strongly correlated substances, as well as its impact on the local magnetic moment formation and screening. As a whole, the topic of the paper, raised problems, and their analysis are certainly very interesting.

Weaknesses

However, I have several suggestions. In particular, I would like to understand better the importance of the triangular vertices lambda_s and other points described below.

Report

The central statement of the paper is that the spin channel provides the most important contribution to the frequency dependence of charge susceptibility, and capable to describe various features of charge susceptibility observed previously in Ref. [22]. To clarify the relative role of vertex corrections and the susceptibility itself, it would be helpful to see in Figs. 3,5,8 the spin contribution with lambda_spin=1. It is not fully clear what is shown by dashed red lines (which are explained as the contribution proportional to chi_s, but not explained in more details). It might be also useful if the authors provide plots of chi_s(T) and/or T*chi_s(T) (possibly in Appendix) to understand evolution of the susceptibilities in different regimes. Is it also a coincidence that the behaviour of different channels in high-temperature and Kondo regime is somewhat similar (although it is different by magnitude)?

The title of the paper "Non-perturbative intertwinning..." looks to me somewhat misleading. Indeed, the authors consider purely perturbative contributions to the charge channel (apart from the irreducible one, which does not play big role in their results). All the non-perturbative information is therefore hidden in the triangular spin vertex and spin susceptibility, which behaviour the authors almost do not analyze. I suggest the authors also to extend the discussion on the non-perturbative aspects in Conclusion and text of the paper.

Another point:
In the beginning of Sect. 3.2 the authors mention formation of relatively flat part of T*chi_s(T) and refer to Refs. [38,49]. However, these references refer to Anderson impurity model, where flat part is absent (T*chi_s monotonously increases). I suggest the authors to cite the papers [22,25] (and possibly others) instead.

Requested changes

  • Add spin contribution with lambda_s=1 to Figs. 3,5,8
  • Explain better the meaning of the dashed red lines
  • Possibly add T*chi_s(T) in Appendix
  • Correct the title and discussion in the text w.r.t. non-perturbative contributions

  • validity: high
  • significance: high
  • originality: high
  • clarity: good
  • formatting: good
  • grammar: perfect

Author:  Severino Adler  on 2023-08-04  [id 3873]

(in reply to Report 1 on 2023-03-11)

We thank the Referee for carefully reading our manuscript, for the positive evaluation of our work and her/his constructive observations.

The Referee has asked us to consider specific points to be addressed prior to publications. We have considered all of them very thoroughly and included the corresponding changes into the manuscript text, figures and the appendix. Specifically, we report below a detailed reply to all the observations of the report. The main questions posed by the Referee are the following ones:

The central statement of the paper is that the spin channel provides the most important contribution to the frequency dependence of charge susceptibility, and capable to describe various features of charge susceptibility observed previously in Ref. [22]. To clarify the relative role of vertex corrections and the susceptibility itself, it would be helpful to see in Figs. 3,5,8 the spin contribution with $\lambda_{spin}=1$.

In order to better clarify the role of the vertex correction to the susceptibility in a coherent way w.r.t.~the flow of the paper, we have included a comparison of the spin contribution to the generalized charge susceptibility obtained with and without approximating $\lambda_{\text{sp}}=1$ in Fig.~12 in Sec.~3.4 and added a corresponding explanation in the text. Panels 2-4 of this figure clearly show the importance of $\lambda_{\text{sp}}$ in the local moment and Kondo regime of the AIM and the HA. The relative role in the local moment regime is an enhancement of the absolute value with respect to the non-interacting limit. This situation is reversed in the Kondo regime, where the spin contribution is largely suppressed in absolute value due to the screening effect of the electronic bath. As described in the text here the Hedin spin-vertex has values smaller than $1$ i.e. its non-interacting limit. For the DMFT solutions in the corresponding two regimes (not shown) the very same considerations apply. To provide a full frequency picture of the effects of this approximation, we have also added the new Appendix C, where a colorplot showing the whole frequency structure (including the frequency off-diagonal elements of the SBE spin contribution) is reported and briefly discussed.

It is not fully clear what is shown by dashed red lines (which are explained as the contribution proportional to $\chi_s$, but not explained in more details).

Regarding the explanation of the red dotted line (note that it was incorrectly indicated as "red dashed" in the main text, which we have corrected in the revised manuscript) in Fig.~3, 5 and 8, we agree that it needs to be extended. Specifically, the spin contribution of Eq.~(11) can be formally split into two parts by inserting Eq.~(10) i.e.

$$ \frac{3}{2} [G_{\nu}]^2\lambda^{\text{sp}}{\nu,\nu'-\nu}\color{green}{(-U+U^2\chi^{\text{sp}}})}\lambda^{\text{sp}{\nu,\nu'-\nu}[G]^2 $$
(note that $\nu'-\nu$ is a bosonic frequency). Keeping only the latter term one gets
$$ \frac{3}{2} [G_{\nu}]^2\lambda^{\text{sp}}{\nu,\nu'-\nu}U^2\chi^{\text{sp}}})\lambda^{\text{sp}{\nu,\nu'-\nu}[G]^2, $$
which is precisely the one we indicated with the red dotted line. This specific SBE contribution, being directly proportional to $\chi^{\text{sp}}(\omega)$ is of most interest for its transparent link to the physical spin response. In order to better clarify this point, we have now added its explicit expression in the caption of Fig.~3 and the main text.

It might be also useful if the authors provide plots of $\chi_s(T)$ and/or $T*\chi_s(T)$ (possibly in Appendix) to understand evolution of the susceptibilities in different regimes.

We agree that such a plot would be useful and have indeed added it together with the new Appendix D. It is also reproduced in the attached structured version of this reply for clarity.

Is it also a coincidence that the behaviour of different channels in high-temperature and Kondo regime is somewhat similar (although it is different by magnitude)?

We assume that the question is mostly referring to the comparison of the data plotted in Fig.~3 and Fig.~8, because only there we show the separate contributions of all channels to the diagonal frequency entries of the generalized charge susceptibilities in the perturbative and in the Kondo screened regime. In this case, we should first emphasize that --on a general level-- one expects that all contributions of our SBE decomposition display larger intensities for lower frequencies along the diagonal, due to their asymptotic decay at high frequency. Further, their specific signs (positive for the bubble, negative for the spin contribution, etc.) appear to be fixed at half-filling, where special particle-hole symmetric properties hold (such as, e.g., that the on-site Matsubara Green's function is purely imaginary). Hence, to a first glance, the structures of the different contributions along the diagonals (unless they are not completely suppressed) might look qualitatively similar. It is also true, however, that beyond this general observation, additional similarities can be noted between the low-frequency perturbative and Kondo regime, due to the screening effects active in the latter case. These are responsible, for instance, of the low-frequency increase (w.r.t.~to the local moment regime) of the bubble term as well as of a moderation of the suppressive contribution of the spin channel, which both drive the (relative) low-$T$ revival of on-site charge response. Obviously, the similarity is not complete. By looking at a more quantitative level, differences also emerge, such as the much smaller/larger contribution of the singlet channel/$U_{\rm irr}$ in the Kondo w.r.t.~to the nonperturbative regime, as well as the almost perfect identification of the spin contribution with its component proportional to the physical susceptibility in the Kondo regime. Eventually, even more evident differences between the perturbative and the Kondo regime can be observed when comparing the off-diagonal frequency structures (e.g., by comparing the third column panels of Fig.~4 and Fig.~9, where the corresponding multiboson contribution is shown).

The title of the paper "Non-perturbative intertwining..." looks to me somewhat misleading. Indeed, the authors consider purely perturbative contributions to the charge channel (apart from the irreducible one, which does not play big role in their results). All the non-perturbative information is therefore hidden in the triangular spin vertex and spin susceptibility, which behaviour the authors almost do not analyze. I suggest the authors also to extend the discussion on the non-perturbative aspects in Conclusion and text of the paper.

This question is of high importance for our work and certainly requires additional clarification (both in the reply and the revised text), as it also touches relevant aspects, which have emerged during the presentation of our results to other colleagues in informal discussions and conferences. Indeed, the Referee is quite right in noticing that one of the pivotal effect we described in the paper, i.e. the sign flip of the diagonal elements of the generalized charge susceptibility is driven by the SBE (and two-particle) reducible scattering processes in the spin channel. In the SBE decomposition, however, no perturbative assumption is -a priori- made, and, as the Referee also noted, the two main constituents of the spin SBE-contribution clearly identified as responsible for the systematic suppression of the diagonal entries of $\tilde{\chi}_c^{\, \nu \nu'}$, namely (i) the (static) physical spin response and (ii) the triangular spin-fermion vertex are the exact ones (for the corresponding case considered) without any a priori restriction to any perturbative approximation. It is important to emphasize, here, that precisely this clear-cut identification via SBE decomposition, which was missing in previous studies (including ours), allows to unveil the physics underlying the breakdown of the self-consistent many-electron perturbation-expansion. In particular, in previous studies, it was just noticed, essentially on a mere empirical basis, that in several fundamental models for strongly correlation, the suppression of on-site charge response occurring the local moment regime of the corresponding phase-diagrams was mostly driven by a strong suppression of the lowest frequencies diagonal entries of the generalized charge susceptibility and, not, e.g., by a generic/uniform reduction of all its matrix elements (which would have been also possible\footnote{For instance this may indeed happen, in the case of a reduction of the density of states of the non-interacting Hamiltonian}. This specific feature is the one determining the breakdown of the self-consistent perturbation expansion, as the suppressed (and then even negative) diagonal entries of $\tilde{\chi_c^{\nu \nu'}}$ causes a sign-flip of its eigenvalues, and, hence, whenever one eigenvalue vanishes, the associated divergences of the irreducible vertex function, the non-invertibility of the corresponding Bethe-Salpeter equation (BSE), and the crossing to physical and unphysical solutions in the Luttinger-Ward functional formalism. A legitimate question posed by many colleagues (as well as by ourself) was then to understand whether the suppression of the on-site charge response associated to a local moment should necessarily occur in this precise fashion (which then unavoidably leads the perturbative breakdown), and, if yes, why this is the case. The identification of the (overall negative!) spin-SBE contribution to $\tilde{\chi}_c^{\, \nu \nu'}$ as the main suppression mechanism of the on-site charge response, presented in this manuscript, has finally provided a clear-cut answer to these questions, in terms of the two main ingredients of the spin-SBE scattering processes mentioned above. Specifically, in the local moment regime (i) the long life-time (actually even infinite in the perfect realization of the local moment, i.e. the Hubbard Atom) of the on-site spin correlations is directly reflected in a selection rule of the major suppression effects of the local charge-fluctuations for $\nu ~ \nu'$ (whereas $\nu \equiv \nu'$ in the "perfect" HA case, where the local spin is a conserved quantity) (ii) the spin-fermion coupling (triangular vertex) gets enhanced w.r.t. its perturbative value of $1$ at low-fermionic frequencies $\nu$. Evidently, the combination of (i) + (ii) explains why the suppression of the on-site charge response, which is unavoidably associated to the formation of a local moment must occur in the precise way observed in the previous work, leading necessarily to a divergence of the irreducible vertex, and, hence, to the breakdown of the self-consistent perturbation expansion. Our analysis, thus, rigorously clarifies the physical nature of the perturbation theory breakdown in all fundamental models considered: The simultaneous enhancement of the on-site magnetic static response and suppression of the on-site charge one, which are both, indeed, intrinsic features of the local moment physics. Hence, any (self-consistent) perturbative approach is bound to fail in describing a proper suppression of the charge fluctuations in the presence of a local magnetic moment, due to the intrinsic impossibility in self-consistent perturbation theory of flipping the sign of any of the eigenvalues of $\tilde{\chi}_c^{\, \nu \nu'}$ , which will remain all positive, as in the corresponding non-interacting case of the model considered. This specific (but relevant!) drawback of self-consistent perturbation approaches has been explicitly observed, e.g. in (truncated) functional renormalization group (fRG) and parquet approximation (PA) calculations, where the local charge response was found to monotonically increase when reducing the temperature even in the local moment regime, reflecting the too weak suppression of $\tilde{\chi}_c^{\, \nu \nu'}$ for $\nu \sim \nu'$ (indeed the diagonal elements of $\tilde{\chi}_c^{\, \nu \nu'}$ remain positive in all fRG and PA dataset). This way, one can eventually understand that the breakdown of the perturbative description in Hubbard model systems is intrinsically rooted into the strong communication between the different physical sectors (magnetic vs. charge, but also particle-particle/pairing), which is essential to yield a self-consistently coherent picture of the local moment physics in its entirety, where the enhancement of the static local spin response must consistently occur together with the suppression on-site charge (and pairing) fluctuations (Note that evidently the same consideration will apply, mutatis mutandis to the formation of local pairs in the case of an attractive on-site interaction (negative $U$)). We note in passing that this strong interplay between the different sectors also represents a crucial ingredient for the (indeed nonperturbative in $U$!) dynamical mean-field theory description of Mott metal-insulator transitions.

We note here -although this is beyond the scope of the present work- that, consistent with our considerations, the unphysical solutions obtained in bold (=self-consistent) diagrammatic Monte Carlo after crossing the first vertex divergence line (i.e. in the nonperturbative regime) are precisely characterized by an unphysical metallicity even in the local moment regime, with a too large charge mobility and even a value of double-occupancy increasing with $U$ (According to several studies bold diagrammatic Monte Carlo resummations do converge also in the nonperturbative regime, albeit not to the correct/physical solution: this is referred to as "misleading convergence" of the self-consistent perturbation expansion, which appears after crossing the first vertex divergence line.).

Another point: In the beginning of Sect. 3.2 the authors mention formation of relatively flat part of $T\chi_s(T)$ and refer to Refs. [38,49]. However, these references refer to Anderson impurity model, where flat part is absent ($T*\chi_s$. monotonously increases). I suggest the authors to cite the papers [22,25] (and possibly others) instead.

Indeed, if one scans the whole temperature range from the high-$T$ perturbative regime down to $T \rightarrow 0$, the quantity $T \chi_s(T)$ for the AIM we considered displays a non monotonous behavior with a rather broad maximum at about $T \leq \frac{U}{2}$. We agree, nonetheless, with the Referee, that our statement about a ``flat part'' of the quantity $T \chi_s(T)$ was rather imprecise and, in general, difficult to be quantified. For that reason, and also in the light of the observation made by the second Referee, in the revised manuscript we have dropped the qualitative statement mentioned above and have refined the corresponding discussion, which also benefited from the additional inclusion of a dedicated figure (showing the behavior of $T \chi_s(T)$ for the HA and the AIM) in the Appendix.

Attachment:

Reply_1.pdf

---

## Round 2 · Referee Report · Anonymous (Referee 2) · 2023-3-14

Report

In the work "Non-perturbative intertwining between spin and charge correlations: A "smoking gun" single-boson-exchange result," S. Adler and co-authors investigate the scattering processes that lead to the development of a fermionic Matsubara frequency structure in the local generalised charge susceptibility in different temperature regimes. The primary finding of this study is to determine the microscopic mechanism that leads to the breakdown of the many-electron perturbation expansion. According to Ref. [1], "the strong suppression of the diagonal entries in \chi^{ch}_{\nu\nu’}, even down to negative values, drives the breakdown of the self-consistent perturbative description, as it is responsible for several sign flips (from positive to negative) of the eigenvalues of the generalized susceptibility and, hence, for corresponding divergences of irreducible vertex functions in the corresponding channel." The authors of the current work demonstrate that this type of frequency structure originates from electronic scattering on the spin susceptibility, which is associated with the formation of a local magnetic moment.

The manuscript presents a very detailed study and is written in a clear manner. However, before I can recommend this work for publication, I would like the authors to address to the following questions:

My main question concerns the interpretation of the results obtained. The authors relate the transition from the high- to intermediate-temperature regime, and consequently the appearance of the specified frequency structure of the generalised charge susceptibility, to the formation of the local magnetic moment. From my point of view this relation is not very well explained and is not well justified.

If this transition is identified by the divergence of the irreducible vertex function in the charge channel, it can only be related to the breakdown of the many-electron perturbation expansion following Refs. [1,3,7,9,10,13-16,18,19] cited by the authors. In Ref. [22], the divergence of the vertex function was associated with the formation of the local magnetic moment, as in the low-temperature and strong-coupling regime the "divergence curve" aligned with the Kondo temperature. However, there was no justification that the divergence of the vertex function could be connected to the formation/destruction of the local magnetic moment in the high-temperature regime.

If the transition is identified by (quoting the authors) "a relative flat (Curie) behavior of the quantity T \chi^{sp}_{ω=0}(T),” then this condition is imprecise. It would be helpful if the authors show the results for the local susceptibility and explain how they identified the transition point. In fact, the deviation from the Curie behavior of the spin susceptibility is rather smooth (see, e.g., [PRB 99, 165134 (2019)]), which usually does not allow one to accurately pinpoint the formation of the local magnetic moment (see, e.g., [arXiv:2112.02881]). In addition, in [PRB 105, 155151 (2022)] it was argued that the formation of the local magnetic moment cannot be captured by the behavior of the static spin susceptibility, because (quoting the authors of that work) the spin susceptibility "cannot distinguish the fluctuations of the local magnetic moment from the spin fluctuations of the itinerant electrons that also contribute to the susceptibility, especially in the paramagnetic regime."

On the contrary, the transition between the low- and intermediate-temperature regimes is clearly defined by the Kondo temperature. Therefore, it would be helpful if the authors specify the values of the Kondo temperature for the systems under consideration and provide an explanation of how they were obtained, as the definition of the Kondo temperature for these systems is not unique. Actually, the issue regarding the formation/destruction of the local magnetic moment is even less clear for the Hubbard atom, which does not have a Kondo regime.

To summarise, it would be helpful if the authors: 1) Provide clear specifications for the transition points between the high-, intermediate-, and low-temperature regimes and elaborate on how these temperatures were calculated. 2) Justify the relation of the mentioned transitions to the formation/destruction of the local magnetic moment, or alternatively, refrain from making such a connection. 3) Demonstrate that the change in the frequency structure is indeed happens at the transition point and not somewhere else in the phase.

In addition, I have two small questions: 1) Could the authors comment on why the charge susceptibility was chosen to study the effect of the formation of the local magnetic moment? If this effect "originates from the electronic scattering on the spin susceptibility," can it be observed directly by examining the spin susceptibility? 2) Could the authors specify the parameters used to obtain the results shown in Fig. 1?

Please correct two typos: 1) Page 3 - “(s. below)” 2) Page 5 - “In order to so”

  • validity: high
  • significance: good
  • originality: good
  • clarity: high
  • formatting: excellent
  • grammar: excellent

Author:  Severino Adler  on 2023-08-04  [id 3874]

(in reply to Report 2 on 2023-03-14)

We thank the Referee for the careful review of our manuscript, for her/his overall positive evaluation, as well as for her/his detailed observations.

Below, we detail our Reply to all specific points raised in her/his report:

The manuscript presents a very detailed study and is written in a clear manner. However, before I can recommend this work for publication, I would like the authors to address to the following questions:

My main question concerns the interpretation of the results obtained. The authors relate the transition from the high- to intermediate-temperature regime, and consequently the appearance of the specified frequency structure of the generalised charge susceptibility, to the formation of the local magnetic moment. From my point of view this relation is not very well explained and is not well justified.

If this transition is identified by the divergence of the irreducible vertex function in the charge channel, it can only be related to the breakdown of the many-electron perturbation expansion following Refs. [1,3,7,9,10,13-16,18,19] cited by the authors. In Ref. [22], the divergence of the vertex function was associated with the formation of the local magnetic moment, as in the low-temperature and strong-coupling regime the "divergence curve" aligned with the Kondo temperature. However, there was no justification that the divergence of the vertex function could be connected to the formation/destruction of the local magnetic moment in the high-temperature regime.

If the transition is identified by (quoting the authors) "a relative flat (Curie) behavior of the quantity $T \chi^{sp}_{\omega=0}(T)$,” then this condition is imprecise. It would be helpful if the authors show the results for the local susceptibility and explain how they identified the transition point. In fact, the deviation from the Curie behavior of the spin susceptibility is rather smooth (see, e.g., [PRB 99, 165134 (2019)]), which usually does not allow one to accurately pinpoint the formation of the local magnetic moment (see, e.g., [arXiv:2112.02881]). In addition, in [PRB 105, 155151 (2022)] it was argued that the formation of the local magnetic moment cannot be captured by the behavior of the static spin susceptibility, because (quoting the authors of that work) the spin susceptibility "cannot distinguish the fluctuations of the local magnetic moment from the spin fluctuations of the itinerant electrons that also contribute to the susceptibility, especially in the paramagnetic regime."

We thank the Referee for this comment. Indeed, the study of the relation between the spin- and the charge-sector in the different regimes (and especially of the local moment one) is one of the central points of our work. Hence, it is important that this aspect is presented in the most clear and convincing way in our reply and in the revised manuscript.

Let us start by stressing (what we have also done in the revised text) that it is not our aim, in this work, to introduce/define or even improve criteria for delimiting the different physical regimes. Indeed, as the Referee points out, this task would be quite hard (if not impossible from a purely rigorous perspective), considering that the different regimes studied in our selected models (HA, AIM, as well as the paramagnetic DMFT solution of the HM on the left side of the MIT) are separated by crossover regions and not by sharp phase transition lines. Our goal is, instead, to precisely rationalize the mechanisms controlling, on the two-particle level, the physics of charge localization, which arguably is "the other side of the coin'' of the local moment formation. In particular, we aim at eventually clarifying how the specific way in which the charge localization gets encoded in the corresponding generalized susceptibilities is linked to the physics underlying the local moment formation and (where applicable) its Kondo screening. In fact (cf.~Introduction and Sec.~II), it was noted in previous works, on an empirical basis, that the freezing of the on-site charge response in the strongly correlated regimes of several basic models was associated to a marked suppression of the diagonal entries of the (corresponding) generalized susceptibility $\chi^{\nu \nu'}$ (which could become quickly negative at low-frequencies) and to a simultaneous slight increase of the off-diagonal elements. However, the essentially empirical nature of such observations prevented to draw rigorous conclusions, leaving the question open (which was posed to some of us several times in conferences and discussions) why the reduction of the on-site charge response driven by correlations was occurring in this precise way rather than, e.g., through an uniform suppression of all matrix elements of $\chi^{\nu \nu'}$, or with a larger suppression of the off-diagonal ones, etc. As the Referee rightly mentioned, this question is also tightly linked to problem of the breakdown of the self-consistent perturbation theory, since the specific (abovementioned) way in which the suppression of on-site charge fluctuations takes place is primarily responsible for the multiple sign-flips of the eigenvalues of $\chi^{\nu \nu'}$, and hence for all the related consequences (divergences of irreducible vertices, crossings of different solutions of the Luttinger-Ward functional, convergence to unphysical results of the many-electron expansion, etc.). Evidently, if the freezing of the on-site charge fluctuations had occurred in a qualitatively different fashion on the two-particle level than the way described before, it might have been possible for self-consistent perturbative approaches to capture the local moment regime physics (including the associated Mott insulating phase in DMFT). We think that our diagrammatic decomposition provides a clear-cut answer to the question above. The obtained results precisely identify the scattering processes (i) mostly responsible for the progressive suppression of the low/inter-mediate diagonal frequency entries of $\chi^{\nu \nu'}$, i.e.~those associated to the electronic scattering with a single spin mode, (ii) as well as those causing the slight overall enhancement of the off-diagonal terms, due to multiple scattering with collective bosonic excitations. It is important to stress that (i) explains then, in a perfectly natural way, why the freezing of on-site charge fluctuation due to the local moment formation happens in the specific way we observe it: The more well-defined the local magnetic moment will be, the longer will be its lifetime (Lifetime, which becomes infinite in the limiting case of the HA, to be regarded, in this sense, as the "perfect gas'' analog for the local moment physics.). In (Matsubara) Fourier space this trend gets immediately reflected in a progressive frequency-localization of the suppressive effects originated by the corresponding single spin-exchange processes on the diagonal entries of $\chi^{\nu \nu'}$. At the same time, this lifetime effect, though crucial, would have not been enough alone to allow for a correct transfer of information between the different channels in the local moment regime. The latter is made possible by the simultaneous low-frequency enhancement of the corresponding spin-fermion scattering amplitude (i.e., of the so-called triangular vertex) w.r.t.~its perturbative/asymptotic value. This diagrammatic identification, which appears numerically quite solid in the three model considered, provides an clear-cut explanations of the question why the freezing of the on-site charge fluctuations happens in the specific nonperturbative manner observed, and how the relevant information (enhanced on-site magnetic response, suppressed on-site charge response) gets transferred between the different physical sectors. In this perspective, in the low-temperature limit of the HA (where the physics of the local moment is essentially perfect, up to vanishingly small exponential corrections of order $\sim e^{- \beta U}$), by estimating the minimal magnitude of the spin-fermion vertex to observe a sign-flip on the diagonal entries of $\chi^{\nu \nu'}$, we were able, finally, to clarify why the size of the frequency-region $[-\nu_{max}, \nu_{\max}]$ where (nonperturbative) negative diagonal values of $\chi^{\nu \nu'}$ are observed scales precisely as $\nu_{max} = \frac{\sqrt{3}}{2} U$. Indeed, the previously empirically determined scaling factor of $\frac{\sqrt{3}}{2}$ finds its most natural explanation in the prefactor of the single spin-exchange contribution, further supporting the validity of our analysis. As we will detail better below, by revising our manuscript we tried to better emphasize the main goal of our study as well as the relevance of the results obtained in this perspective, and to refine/modify imprecise (and, eventually, intrinsically non-conclusive) statements about the borders of the different regimes.

On the contrary, the transition between the low- and intermediate--temperature regimes is clearly defined by the Kondo temperature. Therefore, it would be helpful if the authors specify the values of the Kondo temperature for the systems under consideration and provide an explanation of how they were obtained, as the definition of the Kondo temperature for these systems is not unique. Actually, the issue regarding the formation/destruction of the local magnetic moment is even less clear for the Hubbard atom, which does not have a Kondo regime.

Following the suggestion of the Referee, in the revised manuscript, we have now reported the estimated value of the Kondo temperature ($T_K$) for the AIM considered, as well as of the "effective'' $T_K$ for the DMFT solution of the Hubbard model. The former has been extracted by the temperature dependence of the local magnetic susceptibility $T \chi_m(T)$, namely by matching it to the universal temperature behavior of the Kondo problem, following the procedure detailed in the Appendix A of Phys.~Rev.~B 97 245136 (2018) as well as Sec.~II in Supplemental Material of Phys.~Rev.~Lett. 126, 056403 (2021). We note that this procedure yields, at the $U$ considered, even on a quantitative level the corresponding textbook wide-band limit value for $T_K$ [see, Eq.~(6.109) and ff.~at p.~165–166 of Chap. 6.7 in A.~C.~Hewson, "The Kondo Problem to Heavy Fermions'' (Cambridge University Press, Cambridge, 1993)] (As $T_K$ marks a crossover scale, as the Referee also mentioned, other definitions/criteria could have been chosen. For instance, by estimating $T_K$ via the width of the corresponding Kondo-peak in the spectral function one typically gets larger estimates (up to five times!) than those obtained from the temperature dependence of $T \chi_m(T)$. The latter criterion represents, however, one of the most common choices made in the literature.) , except for the (marginally small!) corrections due to the finite (albeit large) bandwidth of the bath electrons. The precise value of $T_K$ in the energy units of our AIM (Namely, an AIM with a constant DOS of the bath electrons in the interval $[-D,D]$ with $D=10$, hybridization amplitude $V=2$ and impurity on-site interaction $U=5.75$) reads $T_K \simeq \frac{1}{65} \simeq 0.015$. As discussed Phys.~Rev.~Lett. 126, 056403 (2021), in the strong-coupling regime (which applies to the value of $U$ considered here) this (textbook) value of $T_K$ is extremely well approximated by a specific condition on the lowest Matsubara entries of the generalized charge susceptibility, namely that $\chi_c^{\pi T, \pi T} = \chi_c^{\pi T, -\pi T}$. As for the DMFT calculations, where the AIM plays an auxiliary role for the self-consistent determination the corresponding dynamical mean field, the procedure described above cannot be straightforwardly followed, as the auxiliary AIM itself (as well as its Kondo temperature) depends (for a fixed $U$) on the temperature itself. Here, however, by resorting to the criterion based on the lowest Matsubara-frequency mentioned above, one could determine the temperature at which the effective $T_K$ of the corresponding auxiliary AIM is crossed, i.e.~$T_K(T) = T$. For our DMFT calculations on the Bethe lattice, with $U=2.2$ in unit of the half-bandwidth, its estimation yields $T_K \simeq \frac{1}{50}$. Finally, in the Hubbard Atom the local spin operator (as well the local charge operator) is a constant of motion of the problem, allowing to regard this system as an ``ideal realization'' of the local moment physics. The absence of any Kondo screening, yields perfect local moment features in the low-temperature limit, with negligibly small $e^{-\beta U}$ corrections. In that case, the local moment physics can be only be degradated by the thermal activation of the excited states (with $0$ and $2$ electrons, respectively), which occurs at temperatures of the order of the corresponding energy gap, i.e. $T \sim \frac{U}{2}$. Indeed, a brief glance on the temperature dependence of the susceptibilties, allows to appreciate how, in the case of the HA, both the fullfillment of the Curie behavior of the spin response, as well the corresponding exponential suppression of the charge response, become virtually perfect already at (or below) $T \sim 0.5 < \frac{U}{2}$.

To summarise, it would be helpful if the authors: 1) Provide clear specifications for the transition points between the high-, intermediate-, and low-temperature regimes and elaborate on how these temperatures were calculated. 2) Justify the relation of the mentioned transitions to the formation/destruction of the local magnetic moment, or alternatively, refrain from making such a connection. 3) Demonstrate that the change in the frequency structure is indeed happens at the transition point and not somewhere else in the phase.

Considering the questions raised by the Referee, and consistent with our reply, in our revised manuscript we have now refined the presentation of the aims of the paper, underlying that our main goal is not to provide univocal definitions for the crossover borders between the different regimes (which would be, {\sl a priori}, an unfeasible task, due the lack of sharp phase-transitions in the cases considered), but rather to unambiguously identify the scattering processes responsible for the correct intertwining between different physical sectors, i.e. the processes allowing the large local magnetic response due to electronic localization to be accompained (as it should !) by a corresponding suppression of on-site charge fluctuations. We have also emphasized better in the revised version, how this result has allowed us, eventually, to clarify, why such interplay happens in the way we observe it, relating its manifestations to the intrinsic properties of the local moment formation (such as, e.g., to its characteristic long lifetime). In the revised paper, we also provide a concise justification for the choice of parameters (essentially, as $U$ is kept fixed, for the three selected temperatures), we made to study the physics of the different regimes. In particular, for the HA, we've chosen $T = 2 ~ \frac{U}{2} (\beta =0.5)$ for illustrating the behavior of the perturbative regime (of course here, also higher temperatures, but this would have further reduced the "resolution" of our Matsubara susceptibility matrices), and $T=0.1 << \frac U2 (\beta=10)$ for describing the local moment regime. For the AIM (for which we used the same interaction value $U$ as in the HA), the choice of the first two ''higher" temperatures, representative of the perturbative and of the local moment regime is the same as above (as are both much larger than $T_K$), while for the screened (Kondo) regime we selected $T = \frac{1}{60} \sim T_K \simeq \frac{1}{65}$ (compare Fig.~1 of the reply to referee 1). Of course, going along the line of thoughts of the Referee, one might indeed try to exploit the sharper sign-structures characterizing in the generalized charge susceptibility to define new/complementary criteria for delimiting the different regimes of the crossover, similarly to what was done in Phys.~Rev.~Lett. 126, 056403 (2021) for the Kondo Temperature. For instance, for the high-$T$ border $T^{*}(U)$ of the local moment region, one could use the sign-flip from positive to negative of the lowest eigenvalue of the generalized susceptibility, which indeed for large coupling display a scaling with $T^* = \frac{\sqrt{3}}{2}U$. The introduction of such sharp criteria in the charge sector might be even more useful out-of-half filling, where the temperature features in the spin-sector might become even more elusive. At the same time, the introduction of any of such crossover criterion represents (intrinsically) an arbitrary choice, hence, not being the main goal of our study, we prefer not to address explicitly this issue in this study.

In addition, I have two small questions: 1) Could the authors comment on why the charge susceptibility was chosen to study the effect of the formation of the local magnetic moment? If this effect ``originates from the electronic scattering on the spin susceptibility," can it be observed directly by examining the spin susceptibility?

This represents an important point, indeed. On the one hand, as discussed also above, the local moment formation and the associated freezing of local charge fluctuations are the two sides of the same coin ("simul stabunt, simul cadent'', i.e. one cannot have one effect, without the other). Even, the strong intertwining between the two scattering channels (mediated by the enhanced value of the triangular vertex in the local moment regime) is eventually responsible for the breakdown of the perturbation expansion and all its related manifestations. Hence, it would be reasonable to search for the presence of characteristic features of the local moment formation in the generalized magnetic susceptibility, too. On the other hand, the two channels are strongly intertwined, though, they are certainly not equivalent, since the on-site spin response is enhanced and the charge response is frozen. This difference is largely reflected in the corresponding generalized susceptibilities. In particular, one observes that the generalized spin susceptibility displays an enhancement at all frequencies in the local moment regime, whereas the low-temperature Curie behavior of the susceptibility would be associated to a rather featureless positive structure extended on all fermionic Matsubara frequencies $|\nu|,|\nu'| \leq U$. This overall strong, but rather diffuse enhancement makes the fingerprints of the local moment formations, in same sense, not so easy to be directly "read'' from the overall structure of the generalized magnetic susceptibility. On the contrary, the suppression effects on the generalized charge susceptibility (associated to the different signs of its $\uparrow \uparrow$ and $\uparrow \downarrow$ counterparts) is reflected in sharp frequency structures of {\sl different signs} in the charge sector, which are very easy to identify, even at a first glance, and to be directly compared to those observed in other regimes. Beyond this practical reason (a much natural identification procedure), it is also worth to stress here a more general point: The freezing of the on-site charge fluctuations is a crucial aspect in strongly correlated electronic models: It plays an essential role in controlling the electronic mobility properties of the systems. In this respect, we note that precisely these nonperturbative suppressive effects in the charge sector, associated to the formation of the local moments, are responsible for the occurrence of Mott-Hubbard metal-insulator transition in DMFT. Indeed, as mentioned above as well as in Phys.~Rev.~Lett. 126, 056403 (2021), within (even quite advanced) self-consistent perturbation approach, such as truncated fRG and the parquet approximation, even in the presence of local moment features in the spin sector, the freezing of the on-site charge fluctuations does not take place, due to a too little intertwining between the two channels. These considerations further support the choice of focusing on the generalized charge susceptibility.

2) Could the authors specify the parameters used to obtain the results shown in Fig. 1?

We now remarked the parameters in the figure caption.

Please correct two typos: 1) Page 3 - “(s. below)” 2) Page 5 - “In order to so

We thank the referee for the remarks and have corrected the typos.

Attachment:

Reply_2.pdf

---

## Round 3 · Referee Report · Anonymous (Referee 2) · 2023-9-5

Report

I would like to thank to the authors for their efforts in improving the manuscript. Specifically, I appreciate that the authors have incorporated estimates of the Kondo temperature for the Anderson impurity and Hubbard models. However, I have noticed that my initial and, indeed, primary question has not been adequately addressed. It is possible that the authors may not have fully understood my concern, so let me try to rephrase it:

Already in the abstract ("The suppression effect on the diagonal elements directly originates from the electronic scattering on local magnetic moments, reflecting their increasingly longer lifetime as well as their enhanced effective coupling with the electrons") and in the overview of the results (page 8: "the intermediate temperature regime, where a local moment is formed in all the models considered"), the authors assert that the change in the frequency structure of the generalized charge susceptibility between in the high- and intermediate-temperature regimes is linked to the development of the local magnetic moment. However, this assertion lacks proper justification.

In the beginning of Section 3.2, titled "Intermediate-temperature/local moment regime," the authors state: "As the next step, we focus on intermediate temperatures, for which, in the three cases considered, local magnetic moments are formed as a result of the on-site repulsion U. Their formation is signaled by a relatively flat (Curie) behavior of the quantity Tχ sp ω=0 (T) in the temperature range under consideration [39, 48] (not shown, see e.g., Fig. 4 in the supplemental of [23])." First, Figure 4 in the supplemental material of Ref. [23] does not depict the spin susceptibility. Rather, it presents the "partial frequency summation" of the generalized charge susceptibility. This quantity does not provide justification for the formation of the local magnetic moment. Second, in my view, if the authors assert that the change in the frequency structure of the generalized charge susceptibility is related to the formation of the local magnetic moment, they should not only perform calculations in the two regimes where the local moment is either present or absent, but also demonstrate that this change occurs as a consequence of the moment's formation. Otherwise, how can one be certain that this change is not simply linked to the effect of temperature, or to a crossover between incoherent and coherent electronic regimes, or some other factor?

If the authors determine the formation of the local magnetic moment based on "a relatively flat (Curie) behavior of the quantity Tχ sp ω=0 (T)," I strongly urge them to explicitly present the results for the spin susceptibility at various temperatures for all three models considered and to indicate the specific temperature (or a narrow temperature range due to crossover effects) that they associate with the moment's formation.
If the authors further demonstrate that the change in the frequency behavior of the charge susceptibility indeed occurs upon this transition, it would serve as a robust justification.
Alternatively, instead of examining the susceptibilities, the authors could apply the criteria introduced in two recent publications [arXiv:2112.02881 (2021), PRB 105, 155151 (2022)]. These relevant works have been referenced in my questions in the previous review round, yet, for some reason, the authors chose not to address them in their manuscript.

In summary, in my opinion, the connection to the local moment in the current version of the manuscript appears more like wishful thinking than a substantiated justification. Without this justification, the authors could simply refer to the intermediate temperature regime as a regime characterized by the freezing of on-site charge fluctuations without mentioning the local magnetic moment.

As a minor note, there is a typo on page 8, just above Section 3.1, that requires correction: “Fig. ??, s”.
  • validity: high
  • significance: good
  • originality: good
  • clarity: high
  • formatting: excellent
  • grammar: excellent

Author:  Severino Adler  on 2023-11-17  [id 4131]

(in reply to Report 1 on 2023-09-05)
Category:
answer to question

We thank the Referee for her/his second review of our manuscript as well as for her/his overall positive assessment on our work, consistent to the positive evaluation of Referee 2.
The Referee has asked us to consider specific points to be addressed prior to publications. We have thoroughly considered all of them (see the detailed reply below) and included the corresponding changes into the manuscript text and the appendix.

I would like to thank to the authors for their efforts in improving the manuscript. Specifically, I appreciate that the authors have incorporated estimates of the Kondo temperature for the Anderson impurity and Hubbard models.

We thank the Referee for her/his overall appreciation of our revision work.

However, I have noticed that my initial and, indeed, primary question has not been adequately addressed. It is possible that the authors may not have fully understood my concern, so let me try to rephrase it: Already in the abstract ("The suppression effect on the diagonal elements directly originates from the electronic scattering on local magnetic moments, reflecting their increasingly longer lifetime as well as their enhanced effective coupling with the electrons") and in the overview of the results (page 8: "the intermediate temperature regime, where a local moment is formed in all the models considered"), the authors assert that the change in the frequency structure of the generalized charge susceptibility between in the high- and intermediate-temperature regimes is linked to the development of the local magnetic moment. However, this assertion lacks proper justification.

We respectfully disagree with the Referee on this point. As illustrated in our extensive first reply to both Referees, and further discussed below, the link between the local moment physics and the quite specific way in which on-site charge fluctuations are suppressed emerges very clearly from our diagrammatic and numerical analysis. It is possible, however, that part of the misunderstanding on the arguments presented in our previous reply stems from the omitted cancellation of a misleading sentence in the revised manuscript, as we discuss in more detail below.

In the beginning of Section 3.2, titled "Intermediate-temperature/local moment regime," the authors state: "As the next step, we focus on intermediate temperatures, for which, in the three cases considered, local magnetic moments are formed as a result of the on-site repulsion $U$. Their formation is signaled by a relatively flat (Curie) behavior of the quantity $T\chi^{\text{sp}}_{\omega=0}(T)$ in the temperature range under consideration [39, 48] (not shown, see e.g., Fig. 4 in the supplemental of [23])." First, Figure 4 in the supplemental material of Ref. [23] does not depict the spin susceptibility. Rather, it presents the "partial frequency summation" of the generalized charge susceptibility. This quantity does not provide justification for the formation of the local magnetic moment.

We do agree with Referee about her/his critique to the quoted sentence. In fact, as we explicitly wrote in the final part of our first reply, it was our declared intention to remove this sentence by the first revision of the manuscript (Quoting from our previous reply the part referring to this specific sentence in the text: "We agree, nonetheless, with the Referee, that our statement about a "flat part" of the quantity $T \chi_s(T)$ was rather imprecise and, in general, difficult to be quantified. For that reason, and also in the light of the observation made by the second Referee, in the revised manuscript we have dropped the qualitative statement mentioned above and have refined the corresponding discussion, which also benefited from the additional inclusion of a dedicated figure (showing the behavior of $T \chi_s(T)$ for the HA and the AIM) in the Appendix.") . Due to a mere oversight, unfortunately, the sentence was eventually not removed. Thanks to the Referee's comment, in the second revision of our manuscript, we could eventually fix this issue. Though no longer relevant, as the whole sentence has been now dropped, we also acknowledge the incorrect referencing to Fig. 4 (instead of Fig. 1) in the supplemental of [23] (the different figure numbering was referring to the arXiv version of that publication). At the same time, we note that, already in the first revision of the manuscript, we did include, as suggested by both Referees, the full temperature dependence of $T \chi_s(T)$ and $\chi_c(T)$ both for an unscreened (HA) as well as for a screened (AIM) case in final Appendix of our manuscript. The corresponding figures were also supplemented by a brief discussion of the relation between the parameter data sets chosen for the different models and the corresponding regimes.

Second, in my view, if the authors assert that the change in the frequency structure of the generalized charge susceptibility is related to the formation of the local magnetic moment, they should not only perform calculations in the two regimes where the local moment is either present or absent, but also demonstrate that this change occurs as a consequence of the moment's formation.

First, let us note that, the wording "formation of a local moment", though being quite intuitive, lacks a rigorous definition. As we also specify in the revised manuscript, the goal of our paper is not to discuss/compare criteria for delimiting borders between different physical regimes in the models considered. In this respect, let us recall that, as mentioned by the same Referee (as well as in her/his suggested work PRB 105, 155151 (2022), now incorporated in our bibliography), no thermodynamic transition is occurring in any of the systems/parameter regimes we considered. On the contrary, the underlying physics is characterized by smooth crossovers between the different regimes. As such, all possible definitions of the corresponding "borders", more or less recently proposed in the literature, while providing insightful guidelines across the parameter space, cannot be regarded as univocally defined criteria for their intrinsic arbitrariness. For instance, we note that, depending on the criterion adopted, the pretty well established evaluation of Kondo Temperature ($T_K$) does provide estimates which might differ even by a factor of $5$. Hence, consistent with the crossover nature of the physical systems we are considering, the evolution of all quantities (including our physical and generalized susceptibilities) will also be (as it should be) smooth. In this situation, it is quite clear that by presenting additional calculations at the precise parameters, where one or another criterion would set the "border" between two physical regimes, cannot provide insightful information in the specific context of our study (In fact, as we remarked in our previous reply to the second Referee, one could even use the properties of the generalized charge susceptibilities to define additional crossover criteria for these models, similar as the criterion introduced for $T_K$ in PRL 126, 056403 (2021) or to the "fingerprint" criterion tested in PRB 105, 155151 (2022). For instance, one could choose the parameter values where the lowest-frequency diagonal entries of the generalized charge susceptibility drop under a certain value, or become negative, etc. Evidently, this "tautological" choice would not add relevant new information to our analysis. It is worth noticing, however, that such kind of criteria based on the low-frequency property of $\tilde{\chi}^{\rm ch}_{\nu=\nu'}$ would qualitatively (but of course not quantitatively!) match the (different) crossover borders defined in the recent literature. This is now addressed in a footnote at the end of Sec. 2 of the resubmitted manuscript.) . On the contrary, the main goal of our paper is to identify which scattering processes drive the very specific way (s. PRL 126, 056403 (2021)) in which the suppression of the on-site charge response, necessarily associated to the the physics of the formation of a local moment in strongly correlated systems, takes places in all the models considered, and, thus, triggers the breakdown of the self-consistent many-electron expansion. To this aim, we have exploited the Single Boson Exchange (SBE) decomposition, which, differently from parquet-based decompositions, remains applicable in all (perturbative as well as not perturbative) parameter regimes. Our results demonstrate that the SBE-based inspection of the different processes indeed holds the key for a precise understanding of the scattering mechanisms linking the spin to the charge sector, eventually allowing us to clarify the origin of the empirical observations made in PRL 126, 056403 (2021).

In a nutshell, in the parameter regimes where a sufficiently well-defined local moment is present due to electron localization, several among all SBE contributions to the generalized charge susceptibility listed in Eq. (1) of our manuscript (precisely: the charge, the pairing-singlet and the double-counting SBE-terms) become negligible. We note, that while this trend can be clearly seen in our data (cf. Fig. 5 and Fig. 6 in the manuscript), its occurrence is unavoidably tied to the intrinsic physics of the local moment formation and, specifically, to the corresponding suppression of on-site charge and pairing fluctuations. The remaining SBE contributions to the generalized charge susceptibility $\tilde{\chi}^{\rm ch}_{\nu\nu',\omega=0}$, which need to be considered here, are then:

$$ \begin{align} \tilde{\chi}^{\rm ch}{\nu\nu',\omega=0} = &-\beta [G &\ &- [G_{\nu}]^2\phi^{\text{U-irr}}}]^2\delta_{\nu\nu'} &\text{bubble{\nu\nu',\omega=0}[G &\ &+ \frac{3}{2} U^2 [G_{\nu}]^2\lambda^{\rm sp}}]^2 & \text{U-irr{\nu,\nu'-\nu} \, \chi^{\rm sp}(\nu' ! - ! \nu) \, \lambda^{\rm sp}(1) \end{align} $$} [G_{\nu'}]^2 & \text{spin}. & \hspace{1cm

In particular we observed that, while the SBE-irreducible term (U-irr), originated by multiboson exchange processes of all kinds, features a diffuse and overall positive contribution to the generalized charge susceptibility, the only negative (i.e., damping) terms is represented by the scattering processes involving the exchange of a (on-site) spin fluctuation. In particular, as explicitly discussed in our manuscript as well as in our first reply, the more the magnetic moment gets localized, the more its damping effects on the charge sector get concentrated along the diagonal for $\nu=\nu'$. Hence, the link with the physics of the magnetic moment is clear: The less an on-site magnetic moment is screened, the longer will be its lifetime, so that for a perfect\footnote{E.g., where the local magnetic moment becomes a constant of motion of the problem considered.} local moment, one has:

$$ \chi^{\rm sp}(\tau) = \rm{const.} \, \Longrightarrow \, \chi^{\rm sp}(\nu -\nu') \propto \beta \, \delta_{\nu -\nu'}, $$

which corresponds to the Curie-law. The suppressing impact of this increasingly large ($\sim 1/T$) and frequency-selective ($\sim \delta_{\nu -\nu'}$) spin contribution in Eq. (1) gets further enhanced at low-/intermediate frequencies by the the spin-fermion SBE vertex. In the local moment regime this vertex becomes substantially larger (see Fig. 7 in the manuscript) than its perturbative/high-frequency value. This provides a clear explanation of the scattering mechanism driving the suppression of the on-site fluctuations and the smooth emergence of the local moment physics. Remarkably, the precise identification of the link between the suppression of the diagonal entries of $\tilde{\chi}^{\rm ch}_{\nu\nu'}$ and the on-site spin fluctuations in the local moment regime has finally allowed us to rigorously explain the origin of quantitative features characterizing the breakdown of the perturbation theory in the HA, see Eq. (12) in the manuscript, which have been previously reported in the literature, s. PRL 110, 246405 (2013); PRL 114, 156402 (2015); PRB 94, 235108 (2016); PRB 98, 235107 (2018).

Otherwise, how can one be certain that this change is not simply linked to the effect of temperature, or to a crossover between incoherent and coherent electronic regimes, or some other factor?

The direct link between the physical properties of the local moment and the nonperturbative frequency structure of $\tilde{\chi}^{\rm ch}_{\nu\nu'}$ has been very clearly demonstrated by our combined SBE analytical/numerical analysis (cf. our discussion above). To provide further evidence e.g. that the observed behavior of $\tilde{\chi}^{\rm ch}_{\nu\nu'}$ is not driven by temperature dependent single particle coherent/incoherent effects, we have tested our physical interpretation by performing an additional analysis, which we report below. In particular, we have computed how the temperature dependence of the on-site charge fluctuations for the AIM considered in our study would change, if one turned off the specific SBE scattering term associated to the formation of the local moment, while retaining, at the same time, the other $T-$dependent effects associated to the coherence/incoherence of the electronic systems. In practice, this is realized by neglecting all vertex corrections (We note here, that the physics of the local moment formation, as well as the associated low-$T$ enhancement of the corresponding on-site response, is fully driven by vertex corrections. In the absence of the latter, the charge and the spin response would become identical, e.g., displaying both a progressively suppressed behavior at lower $T$ even in models with the strongest local moment effects such as the HA.) to the physical spin susceptibility $\chi^{\rm sp}(\nu -\nu')$ appearing in our SBE expression [Eq. (1) above and Eq. (11) in our manuscript] for $\tilde{\chi}^{\rm ch}_{\nu\nu'}$, i.e. by replacing $\chi^{\rm sp}(\nu -\nu')$ with its corresponding (interacting) bubble term (Keeping the interacting bubble terms allows us to keep the coherent/incoherent $T-$ dependent effects of the electronic system in our calculation. ) $\chi^{\rm sp}_{0}(\omega) = -\frac{2}{\beta} \sum_{\nu} G_{\nu+\omega}G_{\nu} $. The results of our additional analysis are reported in Fig. 1 (visible in the attached reply.pdf file). The data shown in the figure convincingly illustrate the validity of our conclusions: Neglecting the mere vertex corrections included in $\chi^{\rm sp}(\nu -\nu')$ in our SBE decomposition completely cancels any localization effect in the on-site charge response of the system. In fact, this response now displays a monotonous, significant increase when reducing the temperature in the whole $T$-range considered, in spite of the changes occurring in the corresponding on-site Green's function ($G_\nu$). Consistently, by looking at the low-frequency behavior of the generalized charge susceptibility, one can immediately see that also the suppression of the diagonal entries of $\tilde{\chi}^{\rm ch}_{\nu\nu'}$ down to negative values is no longer taking place, thus obliterating one of the main fingerprints of the local moment formation observed in [PRL 126, 056403 (2021)] and discussed in our manuscript.

If the authors determine the formation of the local magnetic moment based on "a relatively flat (Curie) behavior of the quantity $T\chi^{\text{sp}}_{\omega=0}(T)$," I strongly urge them to explicitly present the results for the spin susceptibility at various temperatures for all three models considered and to indicate the specific temperature (or a narrow temperature range due to crossover effects) that they associate with the moment's formation.

As the sentence of our manuscript the Referee is quoting above has been omitted in the newly revised version of the manuscript, we consider this part of her/his observation as resolved.

If the authors further demonstrate that the change in the frequency behavior of the charge susceptibility indeed occurs upon this transition, it would serve as a robust justification. Alternatively, instead of examining the susceptibilities, the authors could apply the criteria introduced in two recent publications [arXiv:2112.02881 (2021), PRB 105, 155151 (2022) to the generalized charge susceptibility ]. These relevant works have been referenced in my questions in the previous review round, yet, for some reason, the authors chose not to address them in their manuscript.

On the basis of the considerations made before in our reply, we think that it would be rather improper to regard any of the different criteria existing in the literature as a marker of a definite "transition", because the local moment formation, as well as its relevant associated effects on the charge sector are smooth crossovers. While the scope of our work is, as mentioned above, a different one, for the sake of completeness of information, we have now included in the revised manuscript a short focused discussion (cf. footnote 3 in Sec. 2 and the extension of Appendix D) of our choice of parameter for the different regimes w.r.t. the crossover borders proposed in PRB 105, 155151 (2022), now Ref. [48], as well as in PRB 105, L081111 (2022). We think that this addition might provide useful information to the readers, and, at the same time, avoid possible misunderstandings.

In summary, in my opinion, the connection to the local moment in the current version of the manuscript appears more like wishful thinking than a substantiated justification. Without this justification, the authors could simply refer to the intermediate temperature regime as a regime characterized by the freezing of on-site charge fluctuations without mentioning the local magnetic moment.

As explained above, our analytical and numerical analysis fully succeeds in directly linking the emergence of the local moment formation (with its intrinsically associated increasing magnitude and lifetime) to the specific way in which the diagonal entries of the generalized charge susceptibility get gradually suppressed, across the corresponding crossover, triggering the breakdown of the self-consistent perturbation expansion. Hence, we hope that with the additional clarifications provided in our reply and revised manuscript, our work can be considered ready for publication.

As a minor note, there is a typo on page 8, just above Section 3.1, that requires correction: “Fig. ??, s”.

We thank the Referee for noticing it: We have fixed the typo in the revised manuscript.

Attachment:

reply.pdf

---

## Round 3 · Author Response

Dear Editor,
Thank you very much for forwarding the two Referee reports on our
manuscript 2212.09693v2:

Non-perturbative intertwining between spin and charge correlations: A "smoking gun" single-boson-exchange result

We are glad that our work has been positively evaluated by the two Referees. In particular, Referee 1 finds that as a whole, "the topic of the paper, raised problems, and their analysis are certainly very interesting", and evaluates as "high" the relevance, the validity and the originality of our study.
Similarly, Referee 2, thinks that our "manuscript presents a very detailed study and is written in a clear manner", rating as "high" the scientific validity of our results, and overall positively their originality and relevance.
Both Referees appear in favor of publication of our manuscript, after we consider their specific
observations to improve the strength of our presentation.

Herewith we resubmit our manuscript carefully revised, after having addresses all the questions raised by the two Referees, as we illustrate in our detailed replies.
Hence, we hope that our revised manuscript can be accepted for publication in Sci. Post. Phys.

Thank you very much for your assistance,

Sincerely yours,
S. Adler, F. Krien, P. Chalupa-Gantner, G. Sangiovanni, and A. Toschi

---

## Round 3 · List of Changes

1.) We added a sentence in Sec. 2.2 to clarify the phase space studied in our DMFT calculations.

2.) Sec. 2.3 was modified to clearly explain the aim and importance of this work.

3.) In the first part of Sec. 3 we added a paragraph to discuss the general features of the crossovers observed in these systems and to specify the reasons for the chosen parameter sets of this work.

4.) In the caption of Fig. 3 the explicit formula describing the red dotted line was added.

5.) An explicit expression for the red dotted line in Fig. 3 was added in the text of Sec. 3.1 and we corrected the word “dashed" to “dotted".

6.) We corrected a typo: we changed the label -5 to -4 in Fig. 3,5,8

7.) We added the new Fig. 12 comparing the diagonal part of the spin contribution to $\chi^{\text{ch}}_{\nu\nu'}$ with and without approximating $\lambda^{\text{sp}}=1$ and a corresponding description at the end of Sec. 3.4.

8.) We modified the text of Sec. 4 highlighting the intent of this paper and, in particular, the nonperturbative character of this study. We also added a short statement about the newly shown results for $\lambda^{\text{sp}}=1$.

9.) We added Appendix C with the new Fig. 13, showing the full frequency structure of the spin contribution under the approximation that $\lambda^{\text{sp}}=1$.

10.) Further, we added Appendix D with the new Fig. 14 showing the evolution of the physical spin and charge response function with temperature.

---

## Round 4 · Referee Report · Anonymous (Referee 3) · 2023-12-18

Strengths

1) Accurate and state-of-the art numerics using various advanced methods 2) Informative and well thought-through plots 3) Interesting question with convincing answer

Weaknesses

1) unnecessarily hard to understand for non-experts since sentences are often too long and convoluted

Report

I have worked through the previous referee reports and the respective answers and in my opinion you, the authors, have addressed the points raised by the previous referees in a satisfactory manner. In particular, the debated formation of a local moment is clearly observed since the local spin operator becomes an (approximate) constant of motion. This is convincingly demonstrated by the dominance of the diagonal contribution in the respective SBE channel.

a) My main concern is about the convoluted and often hard-to-understand way to discuss the results. I urge you to go over the text again, try to sharpen the message and shorten sentences whereever possible. Here is an example from the bottom of page 12:
"At the same time, the low-frequency bubble contribution is, as expected, almost vanishing in the HA, reflecting the ground-state, while it is sizable (though, generally smaller than in the perturbative regime) in the AIM or the DMFT solution of the Hubbard model, consistent with the Fermi-liquid nature of their ground states."
Here is another example from page 5:
"The strong suppression of the diagonal entries in \chi, even down to
negative values, drives the breakdown of the self-consistent perturbative description, as it is responsible for several sign flips (from positive to negative) of the eigenvalues of the generalized susceptibility and, hence, for corresponding divergences of irreducible vertex functions in the corresponding channel [1]. "
Similar constructions appear throughout the manuscript. You need to avoid this type of convoluted sentences if you want to give non-experts a chance to understand your paper.

In the following I list a number of minor points that should be addressed to further improve the readability of the manuscript.

b) On page 14, please explain why you qualify a Green's function with a suppressed low frequency part as "insulating".
c) Please make it very clear where in the manuscript you talk about ONE-line and TWO-line irreducible vertices.
d) You talk about the sign change of the eigenvalues of \chi many times. Why not showing these eigenvalues as a function of T for the models considered if it is so important. Also the need to invert the generalized susceptibility could be explained for completeness. Yes, you may use formulas here!
e) You put yourself in the comfortable position that you only study models for which you can obtain exact correlation functions. But what are the implications of your findings for method development aiming at other more realistic models?
f) There is a typo in Fig. 11 (label "perTurbative") and in the end of the caption of Fig. 14.

Requested changes

see report

  • validity: high
  • significance: high
  • originality: high
  • clarity: good
  • formatting: excellent
  • grammar: excellent

Author:  Severino Adler  on 2024-01-17  [id 4249]

(in reply to Report 1 on 2023-12-18)

We thank the Referee for the careful review of our manuscript, for her/his overall positive evaluation, as well as for her/his constructive observations. Below, we detail our Reply to all specific points raised in her/his report:

a) My main concern is about the convoluted and often hard-to-understand way to discuss the results. I urge you to go over the text again, try to sharpen the message and shorten sentences whereever possible. Here is an example from the bottom of page 12: "At the same time, the low-frequency bubble contribution is, as expected, almost vanishing in the HA, reflecting the ground-state, while it is sizable (though, generally smaller than in the perturbative regime) in the AIM or the DMFT solution of the Hubbard model, consistent with the Fermi-liquid nature of their ground states." Here is another example from page 5: "The strong suppression of the diagonal entries in $\chi$, even down to negative values, drives the breakdown of the self-consistent perturbative description, as it is responsible for several sign flips (from positive to negative) of the eigenvalues of the generalized susceptibility and, hence, for corresponding divergences of irreducible vertex functions in the corresponding channel [1]. " Similar constructions appear throughout the manuscript. You need to avoid this type of convoluted sentences if you want to give non-experts a chance to understand your paper. In the following I list a number of minor points that should be addressed to further improve the readability of the manuscript.

We thank the Referee for this comment. Following her/his suggestion we have now simplified long and convoluted sentences throughout the whole manuscript, improving the readability of our discussions. (s. enclosed .pdf file, with the highlighted changes in the text)

b) On page 14, please explain why you qualify a Green's function with a suppressed low frequency part as "insulating".

For $\text{i}\nu\rightarrow0$ the imaginary Green's function is directly connected to the spectral function at the Fermi level (namely: $-\frac{1}{\pi} G(i\nu \rightarrow 0) = A(\omega=0)$). Therefore, if its value tends to zero, the spectral function at the Fermi level is correspondingly suppressed, featuring an insulating behavior.

c) Please make it very clear where in the manuscript you talk about ONE-line and TWO-line irreducible vertices.

We thank the Referee for his observation: The adjective ``irreducible" was indeed used in two different contexts, though always at the level of two-particle quantities: (i) two-particle vertex functions irreducible w.r.t. a cut of two fermionic lines (mostly in a given channel), which define the typical kernel of parquet/Bethe-Salpeter equations, and (ii) two-particle vertex functions irreducible w.r.t. a cut of a single interaction line ($U$-irreducibility) which naturally appear in the SBE decomposition. Following the observation of the Referee, in the revised manuscript we now always explicitly specify whether we are referring to two-particle irreducible quantities (i.e. to two-particle diagrams that do not fall apart if two fermionic lines are cut) or to U-irreducible quantities (i.e. to two-particle diagrams that do not fall apart when a single interaction line is cut). At the same time, note that we have not used in the paper the word one-particle irreducible for the one-particle quantities, such as the self-energy, as this diagrammatic specification was not needed for our discussions.

d) You talk about the sign change of the eigenvalues of $\chi$ many times. Why not showing these eigenvalues as a function of T for the models considered if it is so important. Also the need to invert the generalized susceptibility could be explained for completeness. Yes, you may use formulas here!

We did not show the results for the sign-changes of the eigenvalues of the generalized charge susceptibility as extensive studies of this specific issue has already been performed in the literature for these models, e.g. in Ref. [1,3,7,9,20,21,23-25,28] mentioned in the Introduction and in Sec. II of the revised manuscript. On the other hand, we agree with the Referee that showing the mathematical expression linking the vanishing eigenvalues of the generalized susceptibility and the divergences of the two-particle vertex functions could definitely help the clarity of our discussions. Hence, in the revised manuscript, namely at p. 14, we have now reported the explicit definition of the two-particle irreducible vertex function, given by the inversion of the corresponding Bethe-Salpeter equation. From that expression, it becomes fully transparent why a zero eigenvalue of a generalized susceptibility triggers a divergence in the two-particle irreducible vertex of the corresponding sector.

e) You put yourself in the comfortable position that you only study models for which you can obtain exact correlation functions. But what are the implications of your findings for method development aiming at other more realistic models?

While the numerical/analytical results of our manuscript rigorously hold for the specific models considered, some general considerations can be plausibly drawn for more general cases, where no (numerically) exact solution is available, and may be used as a guidance for future studies beyond the framework of our work. Indeed, the SBE decomposition is quite general and strong spin fluctuations could manifest themselves in a nonlocal generalized charge susceptibility similarly as in the (purely local) cases studied here. Notice, however, that in the most general case, not only the physical susceptibilities, but also the Hedin vertices are nonlocal quantities. Therefore, it is not easy to predict how local or nonlocal spin fluctuations may be related to corresponding local and/or nonlocal divergences of the two-particle irreducible vertices. These questions are certainly relevant for future investigations, as similar two-particle irreducible vertex divergences in the charge channel have been reported to occur in the cluster DMFT (approximated) solution of the two-dimensional Hubbard model even at lower U-values than those found in the purely local DMFT solution, cf. Refs. [9,16]. Interestingly, these divergences, beyond affecting the lowest frequency structure of the vertex functions, are additionally associated to the specific momentum structure of (commensurate) AF fluctuations (Ref. [16]). One could speculate, thus, that the highly nonperturbative physics of the two-dimensional Hubbard model could be somewhat encoded in a ``generalization" of the strong link between different scattering sectors, which we discuss in our work, beyond the purely on-site physics. In that case, the role of the static local response might be replaced by strong antiferromagnetic and/or RVB fluctuations and could affect, depending on the parameters, the CDW and/or $d-$wave superconduncting response of the systems. Future studies could shed light on these (at the moment purely speculative) ideas.

f) There is a typo in Fig. 11 (label "perTurbative") and in the end of the caption of Fig. 14

We thank the referee for noticing it and have fixed the typo in the revised manuscript.

Attachment:

manuscript_changes.pdf

---

## Round 4 · Referee Report · Anonymous (Referee 4) · 2024-1-6

Report

In this theoretical work the authors investigate the Matsubara structure of the local charge susceptibility for some paradigmatic correlated systems. The goal of this analysis is to identify the microscopic mechanisms controlling, on the two-particle level, the physics of charge localization and its relation with the formation of local moments. To address this issue they exploit the numerical decomposition of the generalized susceptibility in terms of processes involving the exchange of single well-defined bosonic modes. The main result of the study is to unveil the microscopic intertwined between different fluctuating channel. In particular, the study demonstrates that, in the local moment regime, the suppression of local charge fluctuations directly originates from the exchange of static spin fluctuations. The type of analysis performed nicely show that the effect is strongly concentrated for ν = ν ′ as a precise consequence of the long lifetime of the local moment in this parameter regime.

The topic of the research is of high interest for the community working in the wide field of many-body methods and correlated systems. Although highly technical, the paper addresses very intuitive and physical questions. The work performed here, not only provides valuable insight in the physics of correlated material, but also a new strategy to analyze correlation effects at a microscopic level. Such method could be definitely exploited in the future to study more realistic many body models (e.g. including multiorbital physics, non local interactions and more …).

The quality of the research is excellent and the presentation of the results is very good. The organization of the work, the choice of models and parameters and the technical aspects of the computation are well discussed.

I also revised previous referees’ reports and corresponding answers, and I find the reply and relative revision to the paper satisfactory. After consideration of all these aspects, I strongly recommend the paper for publication. In what follows I mention some minor suggestions that I invite the authors to consider:

  1. Some physical aspects, although quite basic could be not obvious for not experts. For example what is the physical meaning of diagonal and not-diagonal component in the Matsubara space in terms of static and dynamic fluctuations, the physical meaning of the matric eigenvalues, as well as the need to invert the generalized susceptibility. The authors could spend a few words in the introductory sections to make the reader able to better appreciate the physics behind the quantities considered.

  2. To help the reader I would use the extensive names of the models in Section 2.2 and define here the acronym used in the following part of the paper.

  3. The paper is already very long and dense. I suggest the authors to have a last revision and try to rephrase/shorten a bit the text whenever is possible.

  4. While the results of the approximations can be briefly mentioned within the discussion to further support the findings, I would suggest to remove “3.4 Insights from Approximations” from the main text and create an appendix.

Requested changes

see report

  • validity: top
  • significance: high
  • originality: top
  • clarity: high
  • formatting: good
  • grammar: good

Author:  Severino Adler  on 2024-01-17  [id 4250]

(in reply to Report 2 on 2024-01-06)

We thank the Referee for reviewing our manuscript as well as for her/his overall positive assessment on our work, consistent to the positive evaluation of Referee 1. The Referee has asked us to consider specific points to be addressed prior to publications. We have thoroughly considered all of them (see the detailed reply below) and included the corresponding changes into the manuscript text and the appendix.

In what follows I mention some minor suggestions that I invite the authors to consider: 1. Some physical aspects, although quite basic could be not obvious for not experts. For example what is the physical meaning of diagonal and not-diagonal component in the Matsubara space in terms of static and dynamic fluctuations, the physical meaning of the matric eigenvalues, as well as the need to invert the generalized susceptibility. The authors could spend a few words in the introductory sections to make the reader able to better appreciate the physics behind the quantities considered.

In the view of our study, some formal aspects, such as the distinction between diagonal and off-diagonal entries of the generalized susceptibilities, are important mostly for their effects in driving the sign-change of the eigenvalues and the associated irreducible vertex divergences (whose link with the structure of the generalized susceptibility have been now more precisely highlighted in the last paragraph of p. 12 of the revised manuscript). Further, as the Referee correctly notes our analysis, based on the SBE decomposition of the generalized susceptibility, also allows to ascribe specific contributions to its diagonal, e.g. those associated to the exchange of a spin collective mode, to the static spin response of the system, which becomes the dominating damping factor of the on-site charge fluctuation in the local moment regime (cf. detailed discussion in Sec. 3.2 and in previous replies). At the same time, one cannot associate, in general, the whole diagonal/off-diagonal entries to one specific static/dynamic response of the systems, because more than one SBE/multiboson term, as well as the bubble-term for the diagonal entries, might contribute to the different matrix diagonal/offdiagonal elements. In this context, in order to highlight the general properties, which characterize the frequency diagonal/off diagonal entries of the generalized charge susceptibility and, in particular, their relation with the bubble and the vertex correction terms, we have slightly extended the discussion right after Eq. (8) in Sec. 2 of the revised manuscript. As for the physical interpretation of the eigenvalues, apart from the suppression effects of on-site fluctuations we mention in our work, which has been extensively discussed, at the level of a significant ``observation'' in the recent literature, see Refs. [3, 25] in the revised manuscript, one should also recall the crucial effect that negative eigenvalues can have in lattice model, when lifting the condition of perfect particle-hole symmetry: The lowest (negative) eigenvalue can trigger the onset of a phase-separation instability, such that observed in the proximity of the Mott metal-insulator transition in the DMFT solution of the Hubbard model. Motivated by the Referee's question, we have inserted an extra sentence at p. 6 before Eq. (8), mentioning this important eigenvalue property and including the specific reference to the papers discussing this aspect (e.g. Ref. [21] and [24]). Eventually, we agree with the Referee that it is also important to explain to the readership, why one might actually need to invert the Bethe-Salpeter equation (and hence the generalized susceptibility matrix). In fact, as we now emphasize in Sec. 2.3 at the end of p. 5 of the revised manuscript, computing such inversions is necessary, e.g., to provide the input for diagrammatic approaches based on parquet equations, such as D$\Gamma$A and QUADRILEX. In fact, at intermediate-to-strong coupling, these approaches might be directly affected by the divergences of the two-particle irreducible vertex functions, triggered by vanishing eigenvalues. In this respect, we also mention in the revised manuscript, that for related reasons, other algorithmic schemes, such as bold diagrammatic Monte Carlo and the nested cluster approaches can face problems in the same parameter regime, as it is discussed in Refs. [2,9].

  1. To help the reader I would use the extensive names of the models in Sec. 2.2 and define here the acronym used in the following part of the paper.

In the light of this Referee suggestion, we now use the full names also in Sec. 2.2 of the revised manuscript and define the acronyms there for a second time.

  1. The paper is already very long and dense. I suggest the authors to have a last revision and try to rephrase/shorten a bit the text whenever is possible.

We thank to Referee for this comment, which has also been mentioned by Referee 1. We have worked throughout the whole manuscript and, whenever possible, we have simplified the text, by shortening and/or rewording too long and complex sentences, without compromising the completeness of information (s. enclosed .pdf file, with the highlighted changes in the text)

  1. While the results of the approximations can be briefly mentioned within the discussion to further support the findings, I would suggest to remove “3.4 Insights from Approximations” from the main text and create an appendix.

We agree with the Referee that the manuscript is long, and can thus benefit from the proposed rearrangement. Hence, we have followed her/his advise by moving Sec. 3.4 into a newly created appendix. In order to keep reference of most important information in the main part of the manuscript we have added a concise paragraph at the end of Sec. 3.3, which summarizes the results obtained via different approximations, with reference to the appendix.

Attachment:

manuscript_changes_5BFxpca.pdf

---

## Round 4 · List of Changes

1. ) Slightly refined the wording of the abstract, for better readability.

2.) Added a footnote in section 2.3 referring to different ways of defining the crossover border of the local moment regime. A more extended explanation has been included at the end of Appendix D.

  1. ) Added a second footnote in section 2.3 referring to the first order jump of the "fingerprint" at the Mott transition.

  2. ) Added reference to PRB 105, 155151 (2022) in various points.

5.) Fixed the typo on page 8 (now on page 9) Fig. ??. It now reads Fig. 14

6.) Dropped the sentence "a relatively flat (Curie) behavior of the quantity $T\chi^{\text{sp}}_{\omega=0}(T)$," at the beginning of Sec.~3.2.

7.) Dropped ", yielding a much less suppressed local moment formation" at the end of Sec.~3.2.

8.) Slightly refined the wording in Sec.~4

9.) Changed $\chi^{\text{sp}}_{0}$ to $\chi^{\text{sp}}_{\omega=0}$ in Appendix B and slightly modified Eq.~(25) and the sentence above.

10.) Changed the scaling of the plots in Fig.~14 to fit the normalization of the rest of the manuscript. Clarified the meaning of the blue shaded area in the corresponding figure caption.

11.) Update/slight extension of the manuscript's bibliography.

12.) Corrected small typos in the manuscript.

---

## Round 5 · Author Response

Thank you very much for forwarding the two Referee reports on our
manuscript 2212.09693v4:

*Non-perturbative intertwining between spin and charge correlations: A "smoking gun" single-boson-exchange result*

We are glad that our work has been positively evaluated by the two Referees. In particular, Referee 1 finds that we *``have addressed the points raised by the previous referees in a satisfactory manner"*, and evaluates the significance of our study as *``high"*. Similarly, Referee 2 thinks that *``the quality of the research is excellent"*, also rating the significance of our study as *``high"* and the validity of our results as *``top"*.

Both Referees appear in favor of publication of our manuscript, after we consider their specific observations aimed at further improving the readability and the impact of our manuscript.

Herewith we resubmit our manuscript carefully revised, after having addresses all the questions raised by the two Referees, as we illustrate in our detailed replies.
Hence, we hope that our revised manuscript can be now accepted for publication in Sci. Post. Phys.

Thank you very much for your assistance,
Sincerely yours,
S. Adler, F. Krien, P. Chalupa-Gantner, G. Sangiovanni, and A. Toschi

---

## Round 5 · List of Changes

1.) Adapted the text improve readability. See attached .pdf to the replies to the Referees.

2.) Used full names of the models in Sec. 2.2.

3.) Added a couple of sentences at the end of the first paragraph of Sec. 2.3 to mention why one might need to invert the generalized susceptibilities and to draw the readers attention to the related problems of diagrammatic approaches, stemming from divergences of the two-particle irreducible charge vertex.

4.) Added a sentence in Sec. 2.3 mentioning the possible physical implications of the sign-change of the eigenvalues, which could not occur in absence of the selective suppression of the diagonal entries of the generalized charge susceptibility.

5.) Added a sentences to highlight the formal connection of the diagonal and offdiagonal elements of $\chi^{\text{ch}}_{\nu\nu'}$ to the ``bubble`` and two-particle vertex correction term in Sec. 2.3 right after Eq. (8).

6.) Included the explicit expression of inversion of BSE as reason for divergences on page 12.

7.) Moved Sec. 3.4 into the appendix, and inserted a concise summary of its content at the end of Sec. 3.3.

8.) Corrected a Typo in Fig. 11.

---

## Editorial Decision

published